# A new method for identifying a fault in T-connected lines based on multiscale S-transform energy entropy and an extreme learning machine

**Hao Wu***, **Jie Yang, Leilei Chen, Qiaomei Wang**

Artificial Intelligence Key Laboratory of Sichuan Province, Sichuan University of Science and Engineering, Zigong, China

* wuhao801212@163.com

## Abstract

Due to the characteristics of T-connection transmission lines, a new method for T-connection transmission lines fault identification based on current reverse travelling wave multiscale S-transformation energy entropy and limit learning machine is proposed. S-transform are implemented on the faulty reverse traveling waves measured by each traveling wave protection unit of the T-connection transmission line, the reverse travelling wave energy entropies under eight different frequencies are respectively calculated, and a T-connection transmission line fault characteristic vector sample set are thus formed. Establish an intelligent fault identification model of extreme learning machines, and use the sample set for training and testing to identify the specific faulty branch of the T-connection transmission line. The simulation results show that the proposed algorithm can accurately and quickly identify the branch where the fault is located on the T-connection transmission line under various operation conditions.

## Introduction

With the continuous development of the social economy, the complexity of the power grid has gradually increased. Considering the investment savings and other restrictions on objective conditions, T-connection transmission lines are increasingly appearing in high-voltage and ultrahigh-voltage power networks due to the uniqueness of their modes of connection. However, these lines are often accompanied by large power plants and systems, whose transmission lines have high transmission power and heavy load. When the line fails, it may cause large-scale blackouts. Therefore, when a fault occurs, it is required to be able to quickly and accurately identify the fault [1–5].

At present, researches on T-connection transmission line fault identification domestic and foreign scholar have carried out are mainly based on the voltage, current and transmission line distribution parameter model. In reference[6], faults occurred within and outside of the protection zone are identified by using the ratio of the phasor sum of T-connection line three-

**Funding:** This research was supported by National Natural Science Foundation of China (Grant Nos.

11705122); The Project of Sichuan provincial science and Technology Department (Grant No. 2017JY0338, 2019YJ0477, 2018GZDZX0043); The artificial intelligence key laboratory of Sichuan province Foundation (2017RYY02); Sichuan University of Science and Engineering talent introduction project (2017RCL53); Enterprise informatization and Internet of things measurement and control technology key laboratory project of Sichuan provincial university (2018WZY01); The Project of Sichuan Provincial Academician (Expert) workstation of Sichuan University of Science and Engineering (2018YSGZZ04).

**Competing interests:** The authors have declared that no competing interests exist.

terminal voltage fault component and the phasor sum of the current fault component. Reference[7] uses the sum of the three-terminal current fault components of the T-connection transmission line and the vector difference between the maximum current in the three-terminal current fault components and the sum of the currents of other two terminals to establish a criterion to identify internal and external faults, but the selection of the braking coefficient in the criterion will have an impact on the sensitivity and reliability of fault identification. Aiming at addressing the problems in reference[7], reference[8] utilized the maximum current in the three-terminal fault current components of the T-connection transmission line with the other two ends and the remaining string angles to establish a criterion to identify internal and external faults, but did not analyze the performance of the algorithm under the influence of noise. Based on the criteria presented in references [7–8], reference [9] establishes a comprehensive criterion to identify faults occurred in the photovoltaic T-connection high voltage distribution network. However, in reference [9], data loss is not discussed in the process of simulation analysis of the algorithm. In reference[10], by using the information of the voltage, current and transmission line positive sequence impedance parameters of each side of the T-connection transmission line, the T-connection voltage is obtained from each side, and then the faulty branch is identified by using the obtained T-connection voltage amplitude information. Reference[11] provides the voltage and current signals measured by the T-connection transmission line protection terminal to the second-order Taylor-Kalman-Fourier (T2KF) filter to estimate the instantaneous values of the voltage and current signals, and then calculates the positive sequence impedance to identify the faulty section. In reference[12], the positive sequence voltage at the T-connection is calculated at the three ends of the T-connection transmission line, and internal and external faults are identified by comparing the maximum amplitude of the T-connection positive sequence voltage superposition component with the maximum amplitude of the three-terminal positive sequence voltage superposition component. In reference[13], the maximum value of the T-connection positive sequence superimposed voltage calculated by the three ends of the T-connection transmission line is used to determine whether the line is faulty, and the phase difference between the positive sequence superimposed voltage and the current at a particular terminal is then used to identify internal and external faults. Reference [14] uses the voltage amplitude difference and measured impedance characteristics of the three sides of the T-connection transmission line to establish the main criterion of the integrated voltage amplitude difference, and with the combination of adaptive distance auxiliary criterion, internal and external faults can be identified. However, the performance of the algorithm has not been simulated. In [15–16], wavelet transform is applied to T-connection transmission line fault identification, but the high-frequency noise signal will affect the identification of faults occurred on T-connection transmission line. In reference[15], the bior3.1 wavelet is used to decompose the three-terminal raw current signal of the T-connection transmission line, the decomposed signal is reconstructed, and then the reconstructed signal is used to calculate the running current and the suppressing current of each phase. Finally, internal and external faults are identified by comparing the relationship between the three-phase corresponding phase running current and the suppressing current. Reference [16] distinguishes internal and external faults faults by comparing the polarity of the fault current detected by the Haar wavelet function at each end of the T-connection transmission line. The fault identification algorithm in reference [17] and [18] is mainly based on the distribution parameters of the T-connection transmission line. Reference [17] discriminates internal and external faults by comparing the exponential sum derived from the model of the transmission line, while reference [18] derives the ranging function based on the distribution parameter model of the transmission line, and uses the phase information at both ends of each branch of the ranging function to determine the branch where the fault is located.

In the traditional fault identification research of T-connection transmission line, the T-connection fault identification algorithm can only identify internal and external faults, but fail to identify the specific branch on which the fault occurred, and the fault identification accuracies in some algorithms are susceptible to other variables. In terms of fault tolerance, the traditional T-connection transmission line fault identification algorithm does not simulate the fault data loss, and cannot verify the fault tolerance of the algorithm. In terms of noise impact, the traditional T-connection transmission line fault identification algorithm does not conduct an in-depth study on it. In order to overcome the shortcomings that the traditional T-connection transmission line fault identification algorithm have in identification precision, accuracy, fault tolerance and effects from noise, this paper further studies the T-connection transmission line fault identification algorithm.

In recent years, S-transformation and information entropy theory have frequently been applied in power systems [19–21]. Reference [22] uses the S-transformed sample entropy ratio of the fault current traveling wave at both ends of the transmission line within a period of time after the fault occurred to identify internal and external faults. Reference [23] established the criterion based on the energy entropy change characteristics obtained by the reverse traveling wave S transform after the faults occurred on each associated line of the busbar to identify internal and external faults.

Based on the theory of directional traveling wave and information entropy expounded in reference [21–23] and with the application of S transform in power system [22–23], this paper proposes a new fault identification method for T-connection transmission line based on the multi-scale S transform energy entropy and limit learning machine of current reverse traveling wave. On the basis of S transformation of fault reverse traveling waves at each end of T-connection transmission line, energy entropy of the reverse traveling waves at 8 different frequencies is calculated to form a sample set of fault characteristic vectors of T-connection transmission line. Combined with the limit learning machine fault intelligent identification model, training and testing are conducted to identify fault branches of T-connection transmission line. Simulation results show that the proposed algorithm can accurately identify the T-connection transmission line branch where the internal or external fault is located under various operating conditions.

## Fault current traveling wave characteristics analysis

### The basic theory of fault traveling waves

Fig 1 shows a 500-KV T connection transmission line. The three branches AO, BO and CO in Fig 1 are defined as the internal branches of the T connection transmission line, and the remaining branches are the external branches.

The connection transmission line is composed of internal branches AO, BO, CO and the external branches AD, BE, and CF. The traveling wave protection units TR1~TR3 are installed at the three ends of the branches near A, B, and C, respectively. When a fault occurs at F1 on branch AO, the traveling wave propagates from the fault point along the transmission line to both sides, and refraction occurs at the discontinuity of the wave impedance of the transmission line. For any point on the line whose distance to the fault point is $x$, the transient voltage and current traveling wave at this point are [24]:

$$\begin{cases} \Delta u(x,t) = \Delta u_+(x - tv) + \Delta u_-(x + tv) \\ \Delta i(x,t) = \Delta i_+(x - tv) + \Delta i_-(x + tv) \\ v = 1/\sqrt{LC} \end{cases} \tag{1}$$

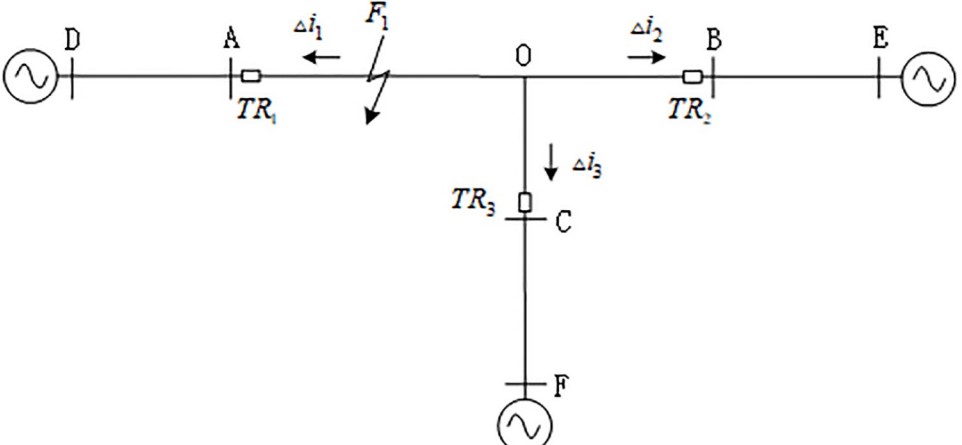

**Fig 1. 500kv T connection transmission line.**

In the equations above, $t$ is the observation time; L and C are the inductance and capacitance of the transmission line per unit length; $\Delta u_+(\Delta u_-)$, $\Delta i_+(\Delta i_-)$ are the voltage and current forward (backward) traveling wave propagating along the positive (negative) direction of $x$.

According to the traveling wave propagation theory, the time at which the initial traveling wave reaches the three ends A, B, and C is $t_{0m}(m = 1, 2, 3)$, respectively, and the second time the traveling wave reaches the three ends of A, B and C after catadioptric reflection occurs is $TR_m(m = 1, 2, 3)$; within the time period $t_{0m} \sim t_{1m}$, the fault traveling wave obtained by the traveling wave protection unit $TR_m(m = 1, 2, 3)$ at the three ends of the branches near A, B, and C is called the initial voltage and the current traveling wave. $\Delta u_m(m = 1, 2, 3)$ is the initial voltage traveling wave measured by the three-terminal traveling wave protection unit of the internal branch near A, B and C, respectively, and $\Delta i_m(m = 1, 2, 3)$ is the initial current traveling wave measured by the three-terminal traveling wave protection unit of the internal branch near A, B, and C, respectively. The wave impedance of the transmission line is $z_c$.

### Analysis of the fault current traveling wave propagation process

**Characteristics of the current traveling wave when an internal fault occurs on the T-connection transmission line.** It can be seen from the analysis that the transient voltage and current at any point of the transmission line are the superposition of forward and backward traveling waves. From Eq (1), it can be concluded that the forward and backward traveling waves of the current are respectively [24]:

$$\begin{cases} \Delta i_+ = \frac{1}{2}\left(\Delta i + {\Delta u}/{z_c}\right) \\ \Delta i_- = \frac{1}{2}\left(\Delta i - {\Delta u}/{z_c}\right) \end{cases} \tag{2}$$

In the equations, $\Delta u$ and $\Delta i$ are the voltage and current fault traveling wave components measured at the point R of each branch; and $Z_c$ is the wave impedance of the transmission line.

It can be seen from the propagation characteristics of the traveling wave that the traveling wave will be deflected at the discontinuity of the transmission line wave impedance (fault point, bus bar, etc.) [24]. According to Fig 1, the positive direction of the traveling wave is defined as the transmission line that the busbar points to. When $F_1$ on internal branch AO of the T-connection transmission line fails, the propagation direction of the current backward

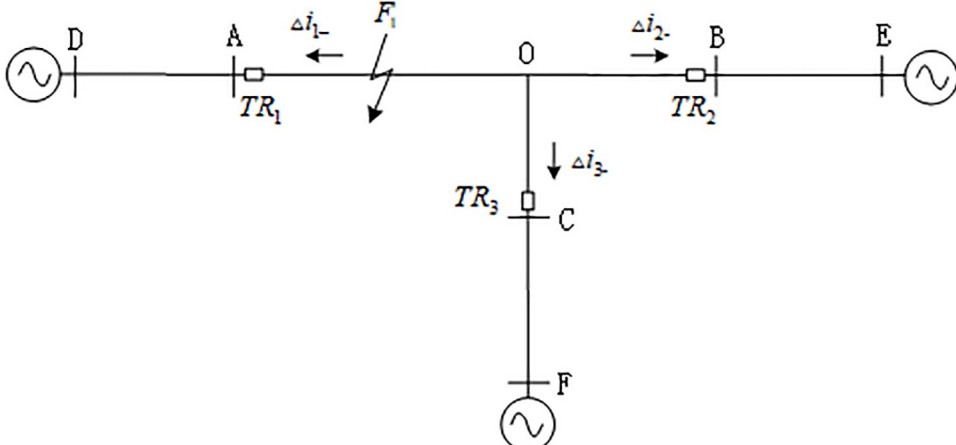

**Fig 2. Propagation direction of the traveling wave when the T-connection transmission line internal branch AO fails.**

travelling wave is as shown in Fig 2, where $\Delta i_{m-}(m = 1, 2, 3)$ is the backward travelling wave of internal branches AO, BO and CO.

When an internal fault occurs on the T-connection transmission line, the traveling wave propagates from the fault point to busbar A, B, and C, and the transmission line and busbar wave impedance are discontinuously deflected. The shortest line within and outside the zone is set to $d_{\min}$, and each traveling wave protection unit $TR_m$ can detect the backward travelling wave within the time period $[t_{0m}, t_{0m} + 2d_{\min}/v](m = 1, 2, 3)$.

**Characteristics of the current traveling wave when an external fault occurs on the T-connection transmission line.**    Fig 3 shows the propagation mode of the directional traveling wave when a fault occurs at $F_2$ on the external branch BE of the T-connection transmission line, and $\Delta i_{m-}(m = 1, 3)$ is the backward traveling wave measured by the protection unit of the internal branch AO and CO, and $\Delta i_{2+}$ is the forward traveling wave measured by the protection unit of branch BO in the zone. During the time period $[t_{0m}, t_{0m} + 2d_{\min}/v](m = 1, 3)$, the protection unit $TR_m(m = 1, 3)$ of the traveling wave can only detect the backward traveling

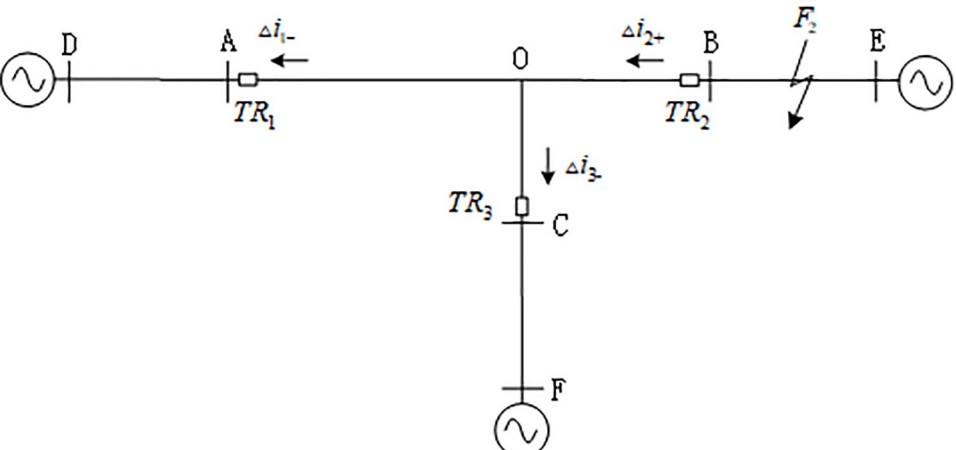

**Fig 3. Fault traveling wave propagation direction when an external fault occurs on the T-connection transmission line.**

wave; within the time period $[t_{02}, t_{02} + 2d_{\min}/v]$, protection unit $TR_2$ of the traveling wave can only detect the forward traveling wave.

## Calculating S-transform energy entropy based on the backward traveling wave

In the three-phase transmission power system, the coupling between the phase voltage and the phase current affects the voltage and current. Therefore, the phase voltage and phase current need to be decoupled. In this paper, the phase voltage and phase current are decoupled with the implementation of Clark phase-mode transformation, and the combined modulus method is used to reflect the various fault types of the T-connection transmission line[25].

$$\begin{cases} \Delta u_z = 4\Delta u_\alpha + \Delta u_\beta \\ \Delta i_z = 4\Delta i_\alpha + \Delta i_\beta \end{cases} \tag{3}$$

In the equations above, $\Delta u_\alpha$ and $\Delta u_\beta$ are Clark's α- and β-modes voltage, respectively; and $\Delta i_\alpha$ and $\Delta i_\beta$ are Clark's α- and β- mode current, respectively.

In this paper, the method used in reference [26] is applied to perform discrete S-transformation on the fault current traveling wave modulus after phase-mode transformation, and the multiscale backward traveling wave energy entropy is calculated by selecting the wave front information of the current backward traveling wave at multiple frequencies after the fault occurs.

### S transform principle

S transform is an extension of the principle of wavelet transform and short-time Fourier transform, which avoids the selection of a window function and break through the limitations of fixed window width. At the same time, the feature quantity extracted by the S transform is not susceptible to noise [27].

Set the continuous time signal as $h(t)$, then the continuous S transformation $S(\tau, f)$ of the time signal $h(t)$ is defined as:

$$S(\tau, f) = \int_{-\infty}^{\infty} h(t)g(\tau - t, f)e^{-i2\pi ft}dt \tag{4}$$

$$g(\tau - t, f) = \frac{|f|}{\sqrt{2\pi}} e^{-\frac{(\tau-t)^2}{2\sigma^2}} \tag{5}$$

In the equations above, $\tau$ is the parameter that controls the position of the Gaussian window on the time axis, $f$ is the continuous frequency, $t$ is the time, $i$ is the imaginary unit, $\sigma = 1/|f|$, and $g(\tau - t, f)$ are Gaussian windows, which are susceptible to the change of frequency.

If $h[kT](k = 0, 1, 2, \cdots, N - 1)$ is a discrete time series obtained by sampling signal $h(t)$, T is the sampling interval, and N is the number of sampling points, then the discrete Fourier transform function of $h[kT]$ is:

$$h[\frac{n}{NT}] = \frac{1}{N}\sum_{k=0}^{N-1} h[kT]e^{-j\frac{2\pi kn}{N}} \tag{6}$$

In the equation, $n = 0, 1, \cdots, N-1$.

Then the discrete S transform of signal $h(t)$ is:

$$S[kT, \frac{n}{NT}] = \sum_{r=0}^{N-1} H(\frac{r+n}{NT}) e^{-\frac{2\pi^2 r^2}{n^2}} e^{j\frac{2\pi rk}{N}}, n \neq 0 \tag{7}$$

$$S[kT, 0] = \frac{1}{N} \sum_{r=0}^{N-1} h(\frac{r}{NT}), n = 0 \tag{8}$$

The complex matrix after the implementation of S transformation reflects the time-domain and frequency-domain characteristics of the signal, as well as the amplitude information and phase information of the traveling wave in the time domain.

## S transform energy entropy

Information entropy is a kind of information measure to the system, which can measure the degree of system disorder, signal uniformity and complexity [20]. The concept of entropy provides a great deal of new ideas for power system fault diagnosis.

Based on the analysis of S transform energy entropy in reference [28], S-transform is applied to the current reverse traveling wave signal $\Delta i_{m\text{-}}(t)$(m = 1, 2, 3) detected by the $m$th traveling wave protection unit. The energy entropy values $W_{mn}$ of time signal sequences at eight different frequencies $f_n$ (n = 1,2,3,4,5,6,7,8) are calculated respectively. The eigenvectors $W_m = [W_{m1} \ W_{m2} \cdots W_{m8}]$ of the energy entropy of the reverse traveling wave at these eight frequencies are defined as Multi-scale S-transform energy entropy vector of signal $\Delta i_{m\text{-}}(t)$.

In this paper, the data of the reverse traveling wave S-transformation energy entropy is calculated by selecting the data within 0.5ms after the fault occurred on the T-connection transmission line (i.e., the data of 50 sampling points before and after the wavefront of the reverse traveling wave). Taking the reverse-traveling wave signal $\Delta i_{m-}(t)$ corresponding to a specific frequency $f_n$ of the m-th traveling wave protection unit $TR_m$ as an example, the calculation steps of the energy entropy are given as follows:

1. Apply S-transform on the reverse traveling wave signal $\Delta i_{mn-}(t)$ and a complex time-frequency matrix is thus obtained, which is denoted as an S matrix. The modulus of each element of S matrix is calculated, and the modulus time-frequency matrix D is obtained.

2. Let the total energy $E_{mn}$ (m = 1, 2, 3) of the signal $\Delta i_{mn-}(t)$ at the specific frequency $f_n$ be equal to the sum of the energy $E_{mn(j)}$ (j = 1, 2, 3,$\cdots$,100) of 50 sampling points before and after the initial traveling wave head of the signal, that is, $E_{mn} = \sum_{j=1}^{100} E_{mn(j)}$,where $E_{mn(j)} = |D_{mn(j)}|^2$ ($D_{mn(j)}$ is the current data of the m-th traveling wave protection unit at the j-th point at frequency $f_n$), and the total energy of the signal is $E = \sum_{m=1}^{3} E_{mn}$. Define $p_{m(j)}$ as the ratio of the energy of the j-th sampling point of the signal to the total energy of the signal., that is, $p_{m(j)} = E_{mn(j)} / E$, then $\sum_{m=1}^{3} \sum_{j=1}^{100} p_{m(j)} = 1$. The m-th traveling wave protection unit single

frequency signal S transform energy entropy is $W_{mn}$, and the multi-scale energy entropy vector is $W_m = [W_{m1}\ W_{m2} \cdots W_{m8}]_{1\times 8}$, in which $W_{mn}$ is defined as:

$$W_{mn} = \left| \sum_{j=1}^{100} p_{m(j)} \log p_{m(j)} \right| \tag{9}$$

### Analysis of reverse traveling waves

**Current reverse traveling waves in the case of an internal fault occurred within T-connection transmission line protection zone.** In the PSCAD model, an A-phase ground fault occurs at the branch AO of the T-connection transmission line protection zone at a distance of 250 km from O. The corresponding waveforms of the current traveling waves of the traveling wave protection units $TR_m (m = 1, 2, 3)$ are shown in Figs 4–6. The signal corresponding to the

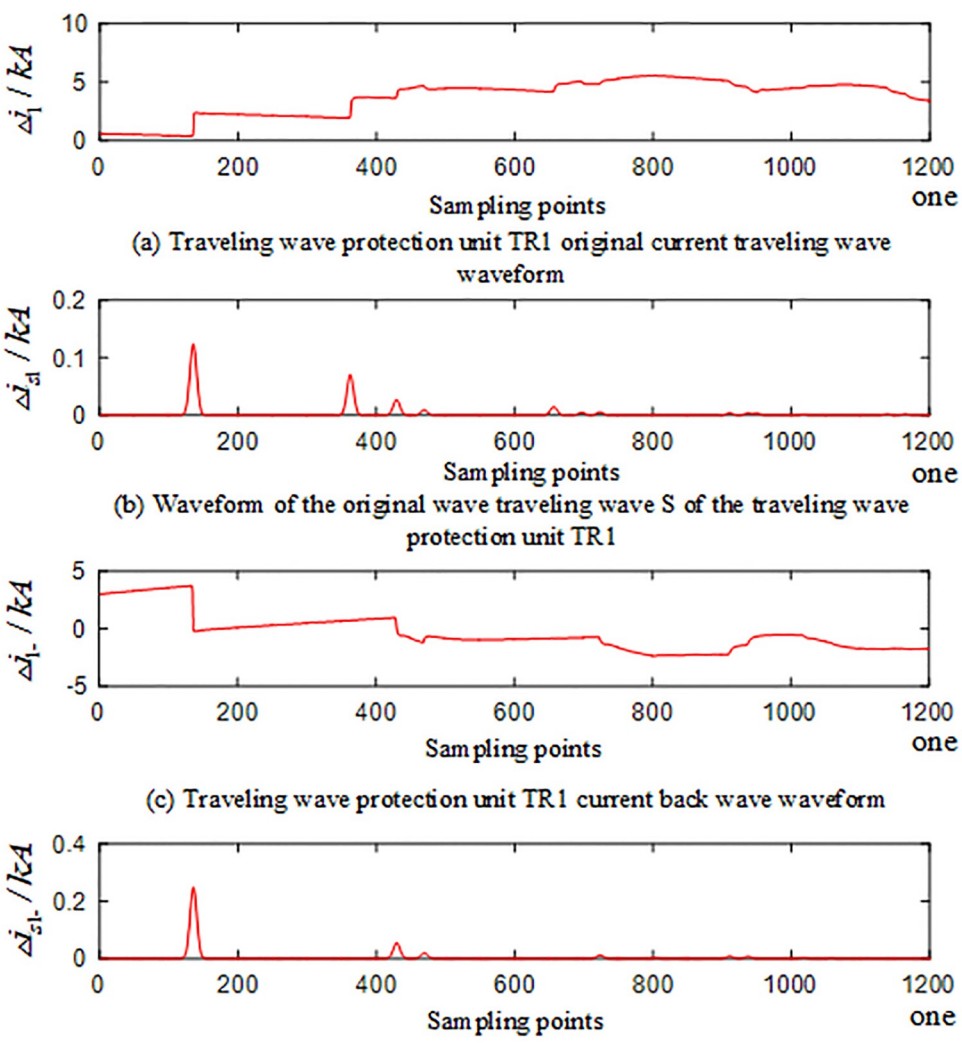

(a) Traveling wave protection unit TR1 original current traveling wave waveform

(b) Waveform of the original wave traveling wave S of the traveling wave protection unit TR1

(c) Traveling wave protection unit TR1 current back wave waveform

(d) Waveform after the traveling wave protection unit TR1 back wave S

**Fig 4. Corresponding waveform of the traveling wave of the traveling wave protection unit $TR_1$ in the case of internal fault occurred on branch AO of the T-connection transmission line.**

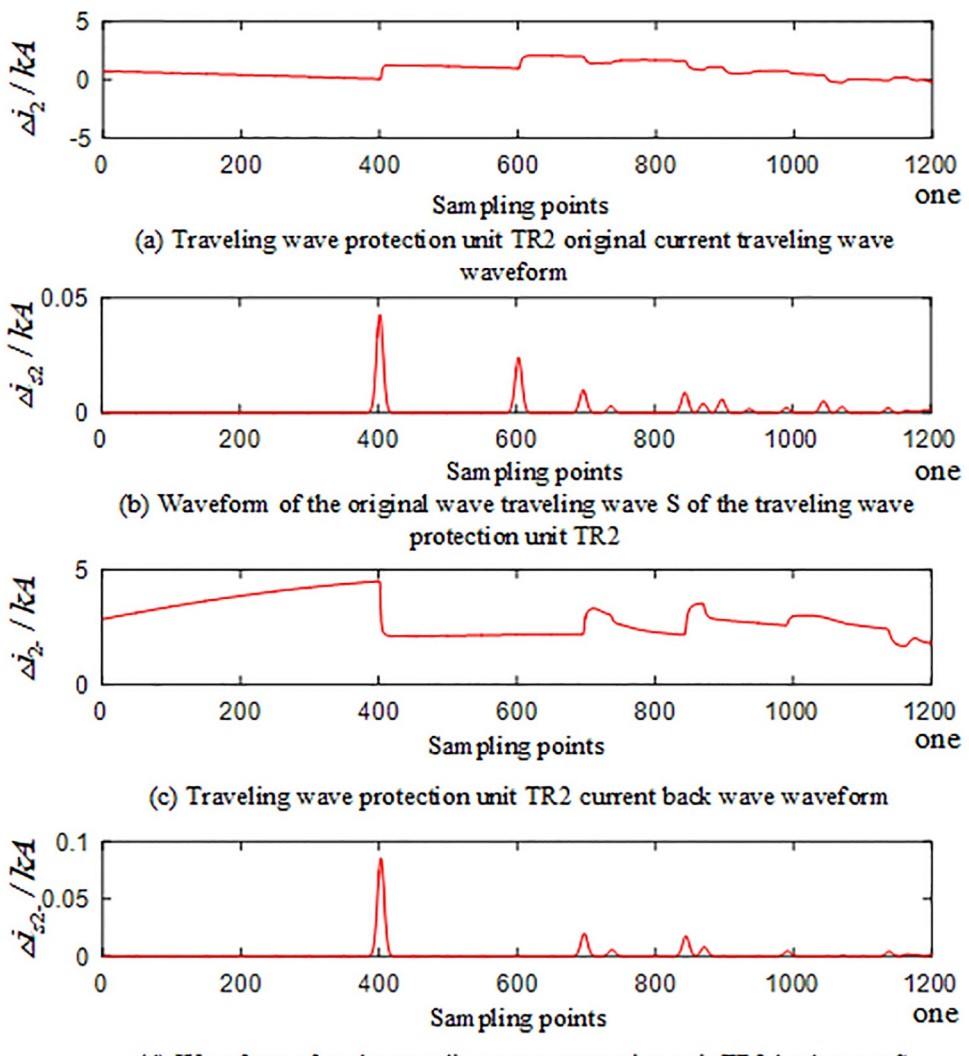

(a) Traveling wave protection unit TR2 original current traveling wave waveform

(b) Waveform of the original wave traveling wave S of the traveling wave protection unit TR2

(c) Traveling wave protection unit TR2 current back wave waveform

(d) Waveform after the traveling wave protection unit TR2 back wave S

**Fig 5. Corresponding waveform of the traveling wave of the traveling wave protection unit $TR_2$ in the case of internal fault occurred on branch AO of the T-connection transmission line.**

40-kHz frequency after the implementation of S transformation is taken as an example, where $\Delta i_m(m = 1, 2, 3)$ represents the corresponding original current traveling wave, and $\Delta i_{m-}(m = 1, 2, 3)$ represents the corresponding current reverse traveling wave.

An analysis of Figs 4 to 6 reveals that when an A-phase ground fault occurs on branch AO within the T-connection transmission line protection zone, the initial current traveling wave measured by each traveling wave protection unit $TR_m(m = 1, 2, 3)$ and the fault current reverse traveling wave appear simultaneously, and the reverse traveling wave can be detected.

**Current reverse traveling wave in the case of an external fault occurred outside the T-connection transmission line.** When an A-phase ground fault occurs on T-connection transmission line branch BE at a distance of 100km from the E-terminal, corresponding waveforms of the current traveling wave of the traveling wave protection units $TR_m(m = 1, 2, 3)$ are shown in Figs 7 to 9. The signal corresponding to the 40-kHz frequency after the implementation of S transformation is taken as an example, where $\Delta i_m(m = 1, 2, 3)$ represents the

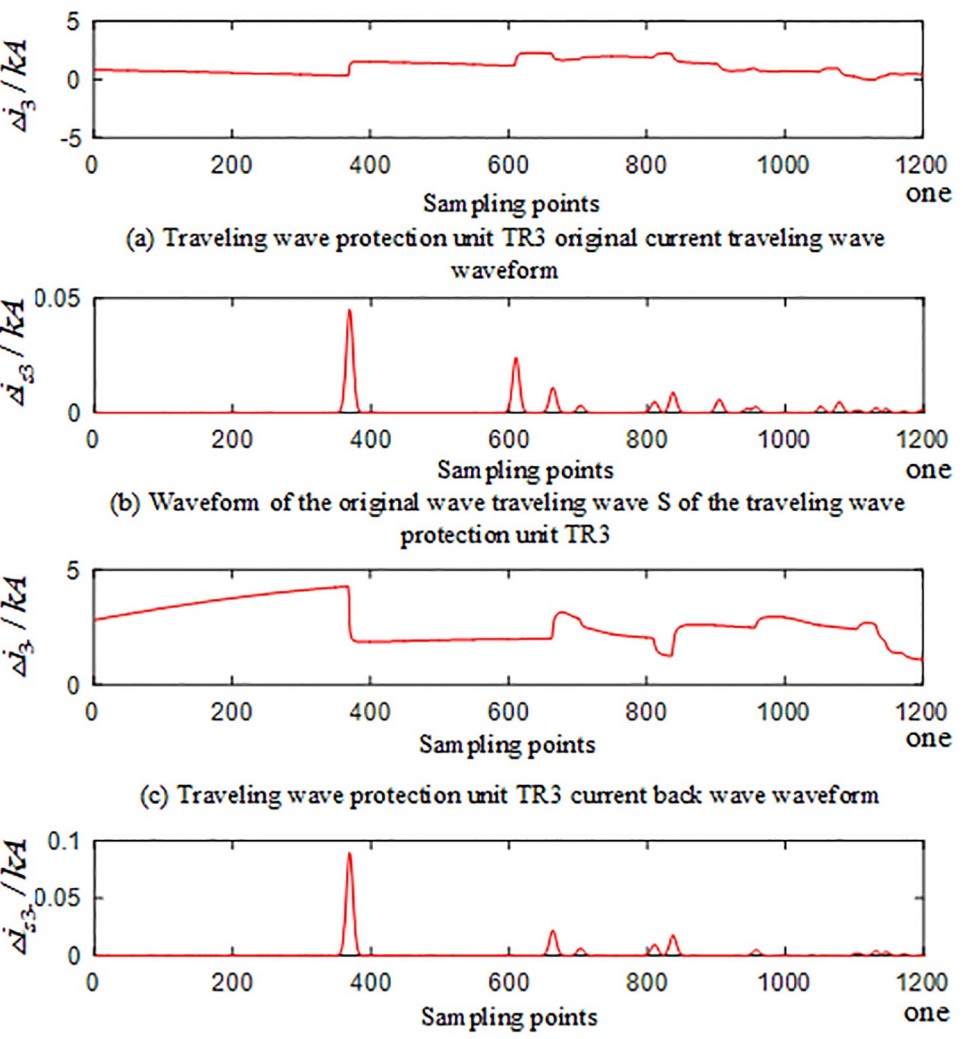

**Fig 6. Corresponding waveform of the traveling wave of the traveling wave protection unit $TR_3$ in the case of an internal fault occurred on branch AO of the T-connection transmission line.**

corresponding original current traveling wave and $\Delta i_{m-}(m = 1, 2, 3)$ represents the corresponding current reverse traveling wave.

Figs 7–9 show that when the external branch BE of the T-connection transmission line fails, the traveling wave protection unit $TR_m(m = 1,3)$ can detect the fault current reverse traveling wave of during the time period $[t_{0m}, t_{0m} + 2d_{min}/v]$ $(m = 1,3)$, while the traveling wave protection unit $TR_2$ can only detect the fault current forward traveling; the current reverse traveling wave cannot be detected.

## Extreme learning machine

Feedforward neural network is one of the artificial neural networks[29]. In this kind of neural network, each neuron starts from the input layer, receives the first input, and inputs to the next level until the output layer. There is no feedback throughout the network, and a directed acyclic graph can be used. Feedforward neural network is the earliest proposed artificial neural

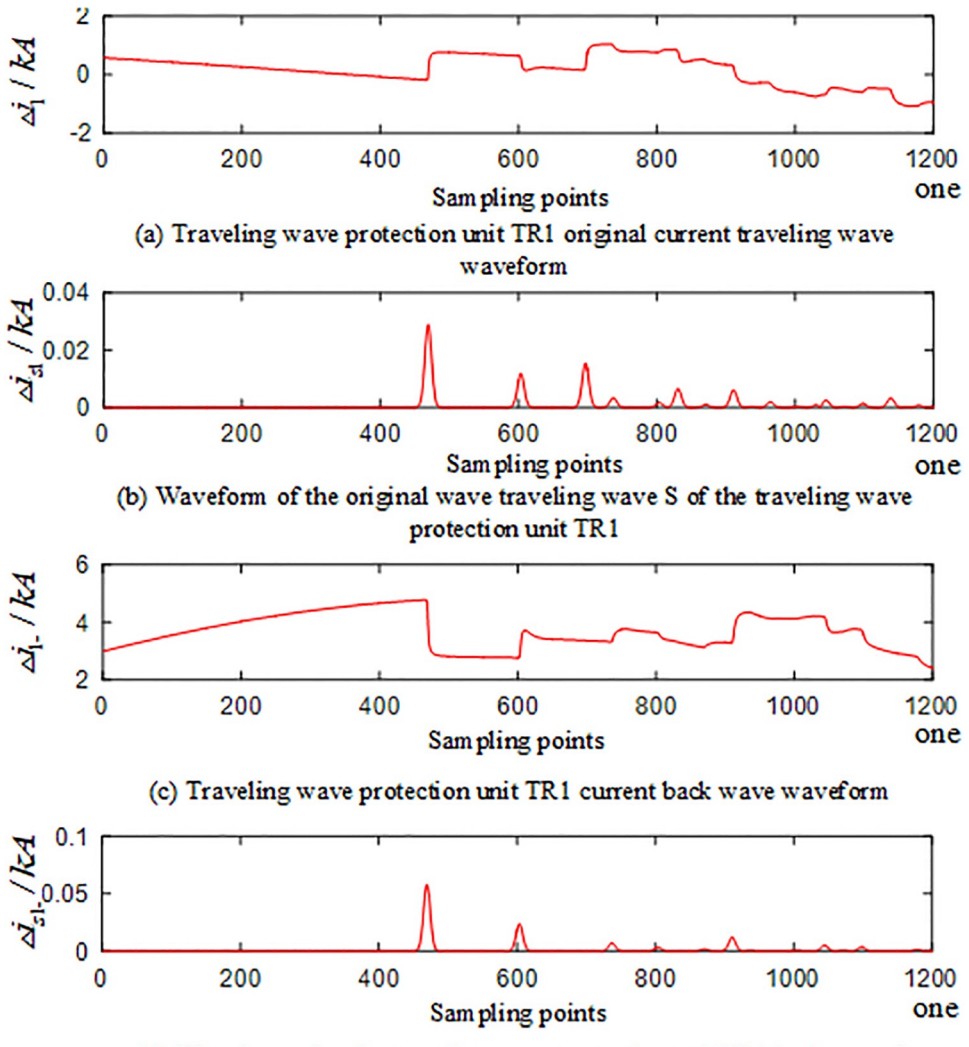

**Fig 7. Corresponding traveling wave waveform of the traveling wave protection unit $TR_1$ in the case of an external fault occurred on branch BE of the T-connection transmission line.**

network and the simplest type of artificial neural network. According to the number of layers of the feedforward neural network, it can be divided into a single layer feedforward neural network and a multilayer feedforward neural network. Among them, common feedforward neural networks include BP neural network[29], radial basis function (RBF) neural network [29] and extreme learning machine (ELM) neural network [30].

An ELM is an easy-to-use and effective single-hidden layer feedforward neural network (SLFN) learning algorithm [30]. The network consists of an input layer, an implicit layer, and an output layer. The neurons of the input layer and the hidden layer, and the neurons of the the hidden layer and the output layer are fully connected. Among them, the input layer has n neurons, corresponding to n input variables; the hidden layer has 1 neuron; the output layer has m neurons, corresponding to m output variables. Fig 10 is a single hidden layer ELM network structure.

ELM only needs to set the number of hidden layer neurons in the network. It does not need to adjust the input weight of the network and the bias of the hidden element during the execution of the algorithm. Compared with the traditional neural network [31–32], it changes the idea that

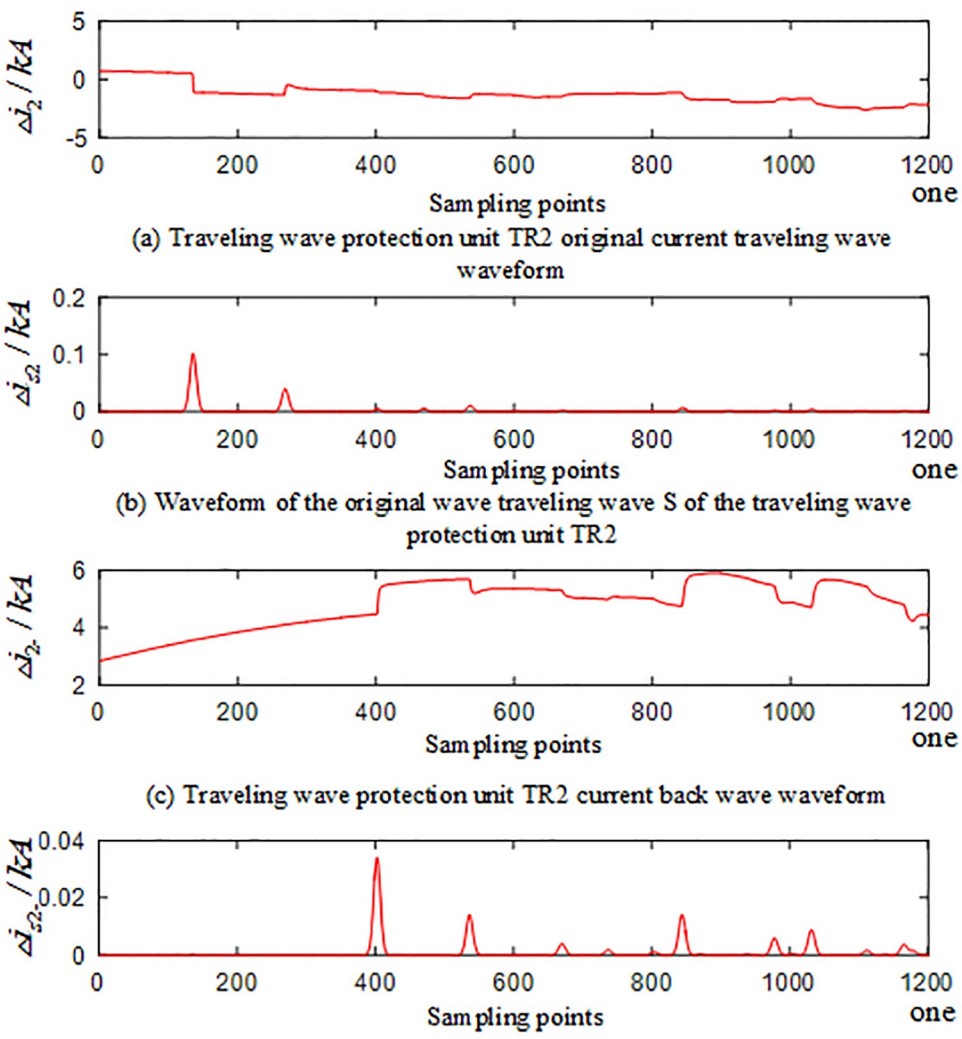

(a) Traveling wave protection unit TR2 original current traveling wave waveform

(b) Waveform of the original wave traveling wave S of the traveling wave protection unit TR2

(c) Traveling wave protection unit TR2 current back wave waveform

(d) Waveform after the traveling wave protection unit TR2 back wave S

**Fig 8. Corresponding traveling wave waveform of the traveling wave protection unit $TR_2$ in the case of an external fault occurred on branch BE of the T-connection transmission line.**

BP neural network should be based on the gradient descent learning and does not need to update the network parameters iteratively. It changes the feature that the learning performance of SVM depends too much on parameter adjustment, and has the advantages of fast learning speed and good generalization performance, and only produces the unique optimal solution.

N different training samples $(x_i, t_i)$ are given, where $x_i = [x_{i1}, x_{i2}, \cdots, x_{in}]^T \in R^n$, $t_i = [t_{i1}, t_{i2}, \cdots, t_{im}]^T \in R^m$, After the excitation function $g(x)$ is given, the output containing L hidden layer nodes can be expressed as:

$$\sum_{i=1}^{L} \beta_i g(a_i \cdot x_j + b_i) = \sum_{i=1}^{L} \beta_i G(a_i, b_i, x_j) = t_i \tag{10}$$

where $j = 1, 2,\ldots,N$; $a_i = [a_{i1}, a_{i2}, \cdots, a_{in}]^T$ is the input weight of the input node and the i-th hidden layer node; $b_i$ is the neuron offset of the i-th hidden layer node; and $\beta_i = [\beta_{i1}, \beta_{i2}, \cdots, \beta_{im}]^T$ is the output weight of the i-th hidden layer node and the output node.

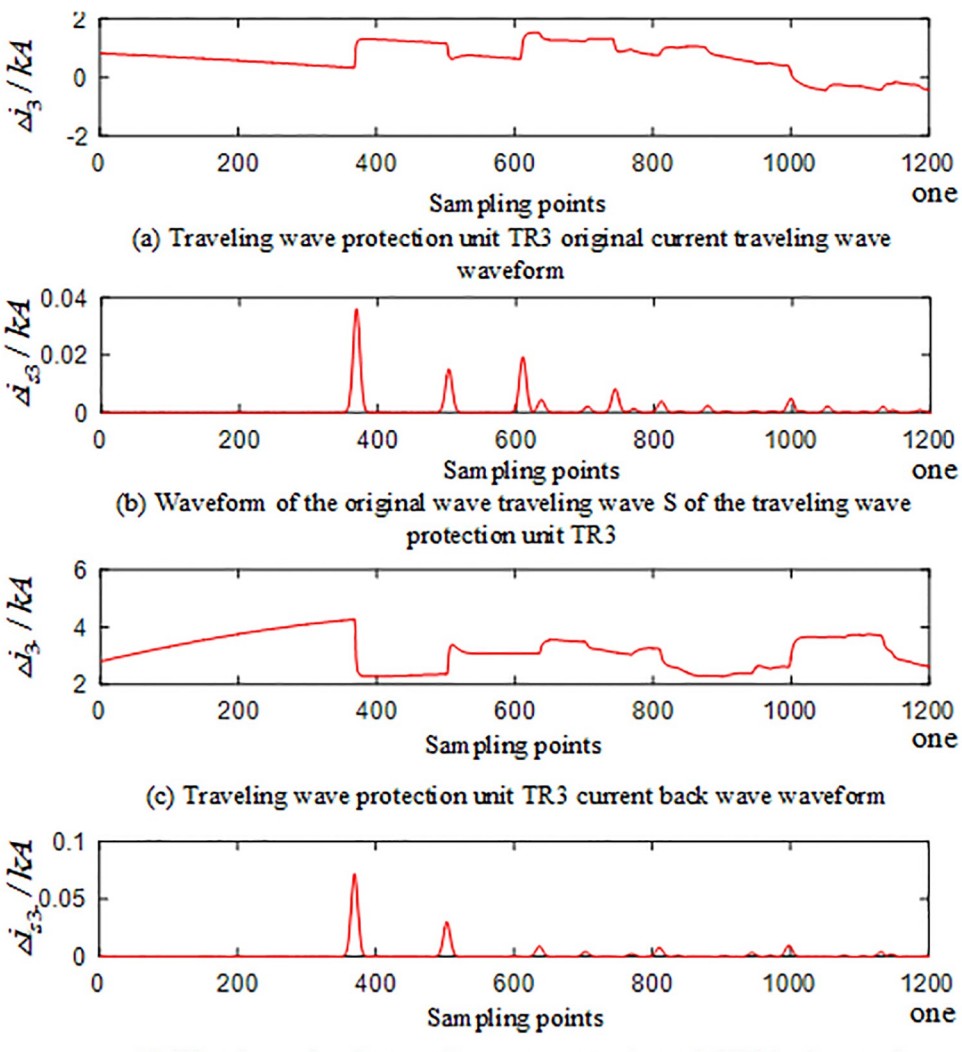

**Fig 9. Corresponding traveling wave waveform of the traveling wave protection unit $TR_3$ in the case of an external fault occurred on branch BE of the T-connection transmission line.**

The algorithm steps are as follows:

1. Randomly select $(a_i, b_i)$ and map the sample to the new feature space through $h(x) = [G(a_1, b_1, x), \cdots, G(a_L, b_L, x)]^T$. Random feature mapping $h(x)$ forms the hidden layer matrix $H$; then,

$$H\beta = T \tag{11}$$

where $H = \begin{bmatrix} h(x_1) \\ \vdots \\ h(x_N) \end{bmatrix} = \begin{bmatrix} G(a_1, b_1, x_1) & \cdots & G(a_L, b_L, x_1) \\ \vdots & \vdots & \vdots \\ G(a_1, b_1, x_N) & \cdots & G(a_L, b_L, x_N) \end{bmatrix}_{N \times L}$, $\beta = \begin{bmatrix} \beta_1^T \\ \vdots \\ \beta_L^T \end{bmatrix}_{L \times m}$, and

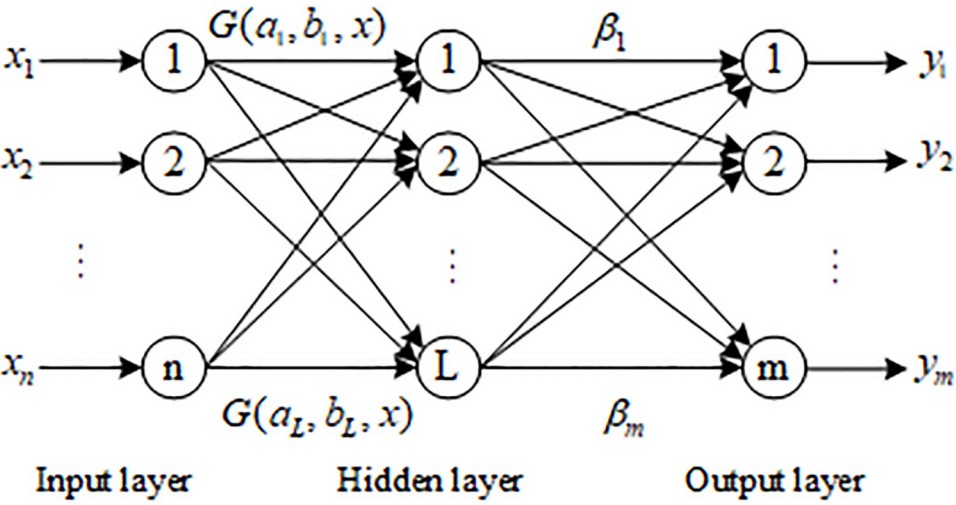

**Fig 10. ELM network structure.**

$$T = \begin{bmatrix} t_1^T \\ \vdots \\ t_N^T \end{bmatrix}_{N \times m}$$ . The implicit layer node excitation function selects the sigmoid function:

$$g(a_i \cdot x_j + b_i) = \frac{1}{1 + \exp(-a_i \cdot x_j + b_i)}.$$

2. In the new feature space, as specified in Eq (11), the least square method is used to calculate the optimal output weight $\hat{\beta}$, where $\hat{\beta} = H^+ T$, and $H^+$ is the Moore-penrose generalized inverse of $H$.

## New T-connected transmission line fault identification method

S-transform is implemented on the traveling wave data detected by the traveling wave protection unit $TR_m(m = 1, 2, 3)$ of T-connection transmission line after fault occurs, and the current reverse traveling wave data corresponding to 5, 10, 15, 20, 25, 30, 35 and 40 kHz of the traveling wave protection unit after the implementation of S-transformation are selected. The energy entropy vector of multi-scale reverse traveling wave $W_m$ is constructed by calculating the reverse traveling wave energy entropy at different frequencies (take the data within 0.5ms after the fault, corresponding to 100 data points), and $W_m = [W_{m1}\ W_{m2} \cdots W_{m8}]_{1 \times 8}$. The energy entropy vector of multi-scale reverse traveling wave of three traveling wave protection units is combined into a fault eigenvector W of T-connection transmission line to reflect the feature of the fault occured on the branch of T-connection transmission line. The fault branch is labeled as the sample data of the extreme learning machine, where $W = [W_{11} \cdots W_{18}\ W_{21} \cdots W_{28}\ W_{31} \cdots W_{38}]_{1 \times 24}$. The fault identification algorithm flow is shown in Fig 11.

## Simulation and experiments

The PSCAD/EMTDC electromagnetic transient simulation software is used to establish a 500kV T-connection transmission line simulation model shown in Fig 12. The model adopts the frequency dependent distribution parameter model that can accurately reflect harmonic and transient responses. TOWER:3H5 pole tower is selected as the type of the line. The

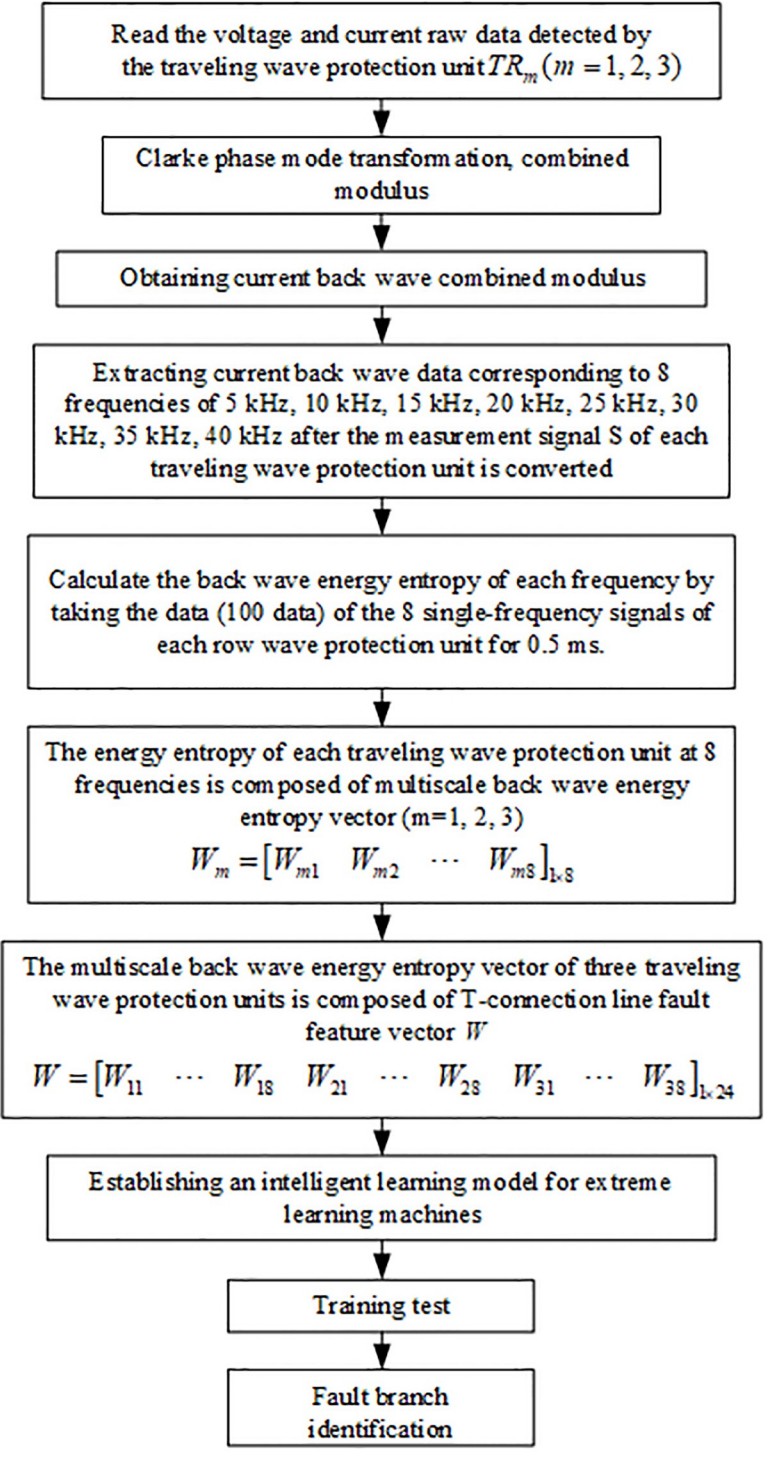

**Fig 11. Flow chart of the fault branch identification algorithm.**

configuration of the power transmission line is shown in Fig 12 below. The parameters of the transmission line are shown in Tables 1 and 2 below. The simulation sampling frequency is 200 kHz, and the length of each branch is AO = 300 km, BO = 200 km, CO = 150 km, AD = 170 km, BE = 150 km, CF = 180 km, respectively. The current reverse traveling wave

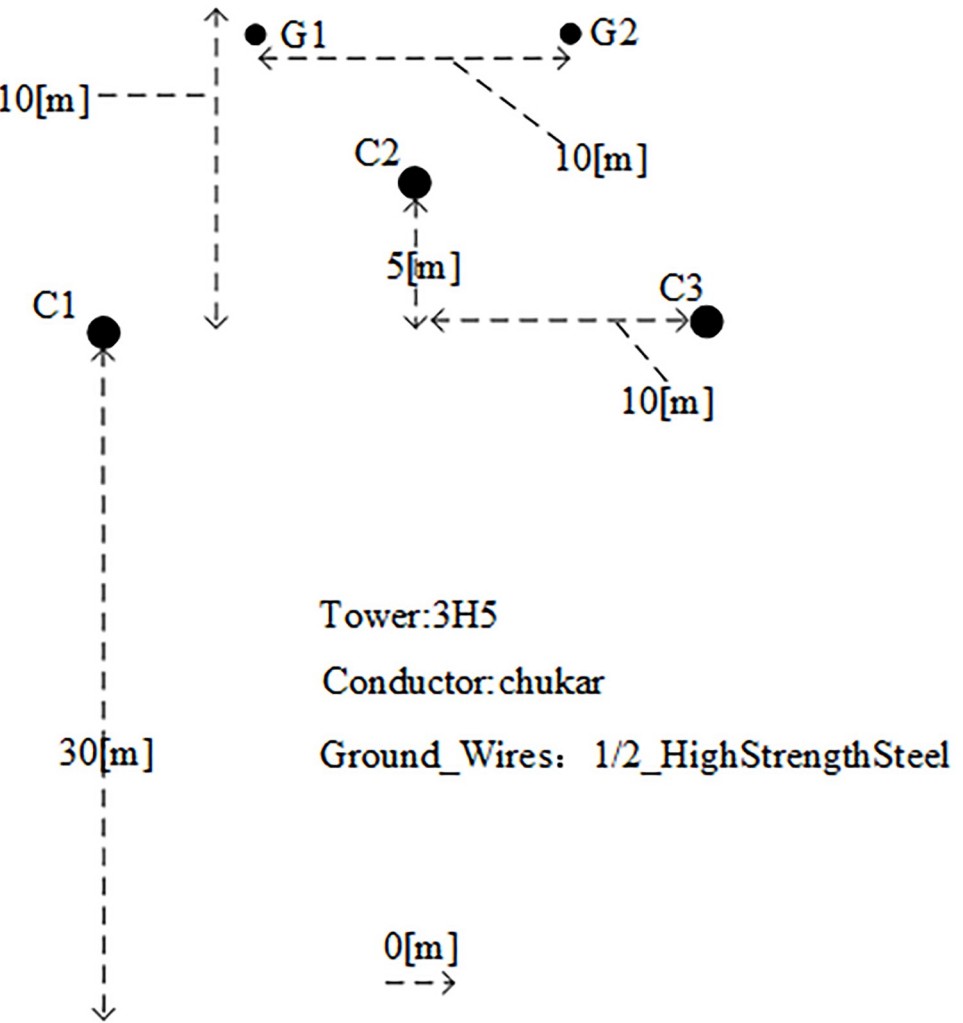

**Fig 12. Power line configuration.**

data corresponding to 5, 10, 15, 20, 25, 30, 35 and 40 kHz of the traveling wave protection unit after the implementation of S-transformation are selected in order to calculate the reverse traveling wave energy entropy of each frequency. The energy entropy vector of multi-scale reverse traveling wave $W_m = [W_{m1}\ W_{m2} \cdots W_{m8}]_{1\times8}$ is constructed. The energy entropy vector of multi-scale reverse traveling wave of three traveling wave protection units is combined into a T-connection transmission line fault eigenvector W to reflect the feature of the fault occurred on the branch of the T-connection transmission line. The sample data of the extreme learning machine is constructed, where $W = [W_{11} \cdots W_{18}\ W_{21} \cdots W_{28}\ W_{31} \cdots W_{38}]_{1\times24}$.

**Table 1. Transmission line parameter I.**

| Type of the Line | Parameter | numerical value |
|:---:|:---:|:---:|
| **Phase line** | Wire radius/m | 0.0203454 |
| | DC Resistor/(Ω/km) | 0.03206 |
| **Ground wire** | Wire radius/m | 0.0055245 |
| | DC Resistor/(Ω/km) | 2.8645 |

**Table 2. Transmission line parameters II.**

| | Resistance R(Ω/km) | Reactance X(Ω/km) | Conductance G(s/km) | Senator B(s/km) | Capacitance C(μF/km) |
|---|---|---|---|---|---|
| **POS** | 0.0346755486 | 0.423365555 | 0.0000001 | 0.00000272598288 | 0.0135 |
| **ZERO** | 0.30002296 | 1.1426412 | 0.0000001 | 0.000193555082 | 0.0092 |

## Sample data

To verify the validity and reliability of the algorithm, this paper chooses to simulate the branches within and outside the T-connection transmission line protection area under different fault types, different transitional resistances, different fault distances and different initial angles of faults.

The training sample of the extreme learning machine consists of two parts: samples with no missing sampled data and sample after the sampled data is partially lost when the branches of the T-connection transmission line fail. Samples with no missing sampled data is composed of sample of a random fault in the branch and Near-O point fault sample within the T-connection transmission line protection area. The random fault samples of the branch are 120 sets of fault eigenvectors obtained by simulating five different faults occurred on six branches of the T-connection transmission line under different fault conditions; The near-O-point fault sample within the T-connection transmission line protection area are 30 sets of fault eigenvectors obtained by simulation under different fault conditions, obtaining fault distance of 5km, 4.5km, 4km, 3.5km, 3km, 2.5km, 2km, 1.5km, 1.2km, 1km from point o on branch AO, BO, CO. Sample after the sampled data is partially lost are 60 sets of fault characteristic vectors obtained by losing 10, 20, 30, 40, and 50 sample points of data near the wavefront of the current traveling wave under selected 8 frequencies. T-line fault characteristic vector with and without data lost together constitute a fault characteristic training sample set, which is input into the limit learning machine for training. The training sample set is composed as shown in Fig 13 below.

The data of 6 branches of T connection transmission line different from the fault types of training samples are taken as test samples respectively, which are input into the trained intelligent fault identification model of the limit learning machine, and the fault branches are identified and tested.

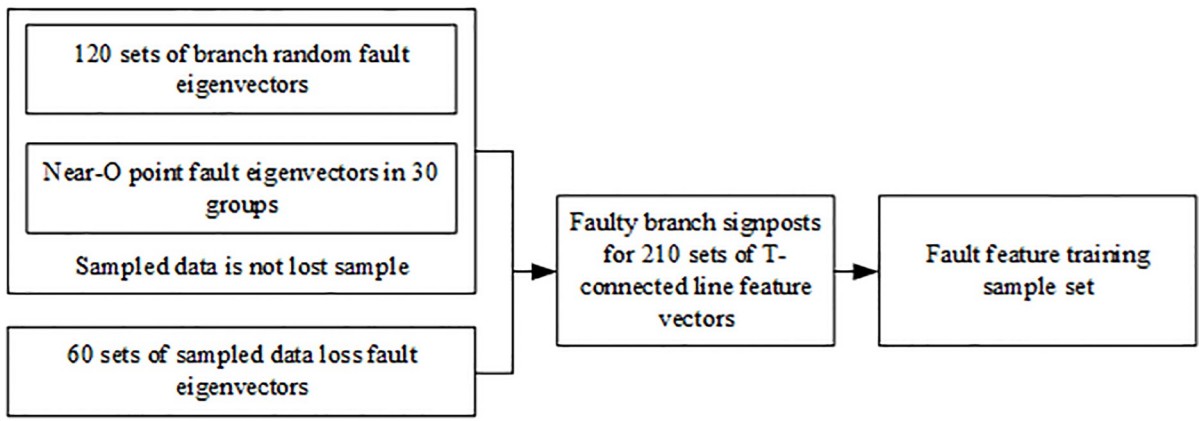

**Fig 13. Training sample set composition diagram.**

### Extreme learning machine intelligent fault identification model establishment and training sample test analysis

The fault characteristic training sample is input into the limit learning machine for training, and a trained extreme learning machine T-connection transmission line intelligent fault identification model is obtained. The optimal number of neurons in the hidden layer of the extreme learning machine obtained by the trial and error method is 70.

The fault characteristic training samples are input into the trained extreme learning machine intelligent fault identification model for testing, and the comparison of the predicted results is shown in Fig 14.

The above figure shows that the test sample data in the ELM fault intelligent identification model have a correct rate of 100%.

### Test sample test analysis

The fault characteristic test samples with different fault types, different transitional resistances, different fault distances and different initial fault angles are input into the T-connection transmission line intelligent fault identification model to identify the faulty branch, and the test results of the test sample set are analyzed.

**Analysis of different fault type tests.** The fault characteristic test samples of different fault types within and outside the protection zone are input into the T-connection transmission line intelligent fault identification model of the limit learning machine for testing. A comparison of the prediction results is shown in Fig 15, and the simulation verification results corresponding to the fault conditions are reported in Table 3.

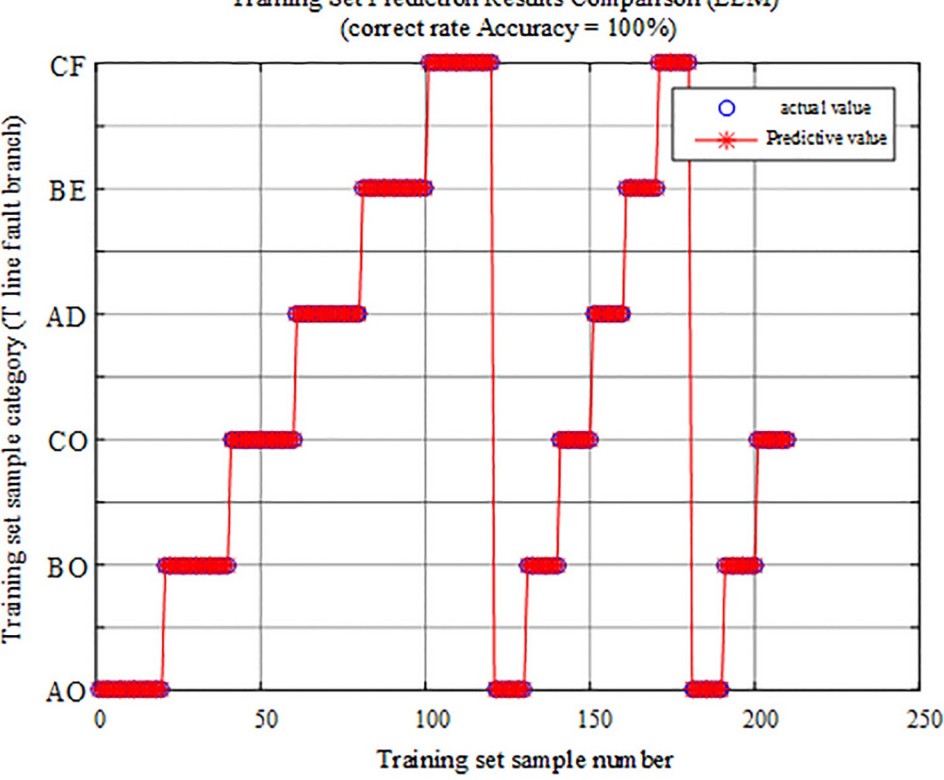

**Fig 14. Comparison of training set prediction results.**

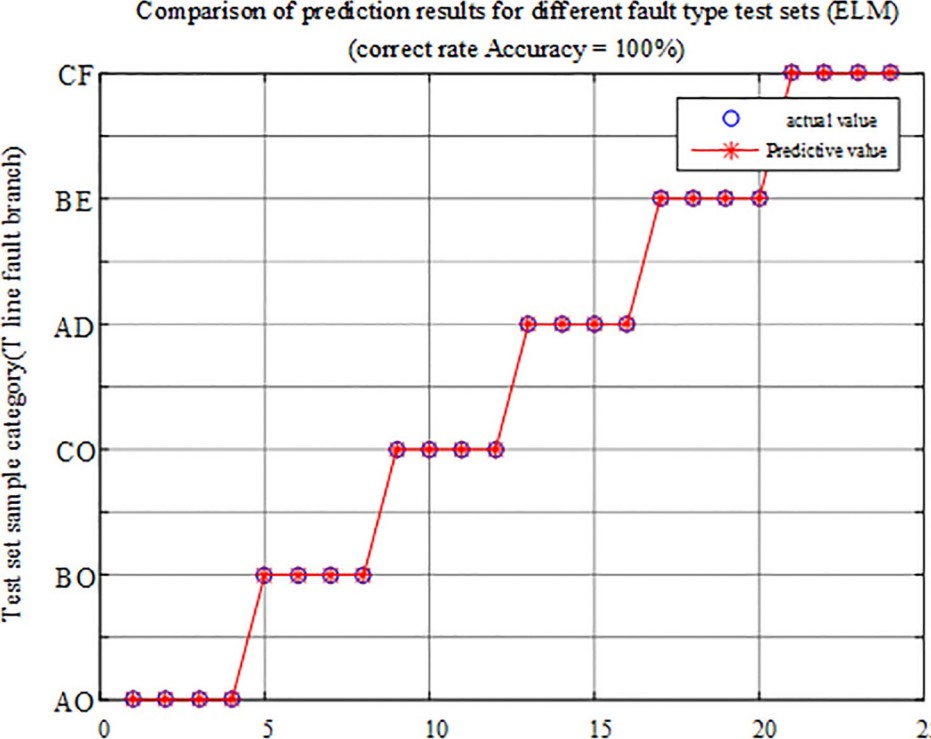

**Fig 15. Comparison of test set prediction results.**

**Table 3. Simulation results of different fault type test sets.**

| Fault branch | Fault type | Fault initial angle/degree | Fault distance O point / km | Transitional resistance / Ω | identification result |
|---|---|---|---|---|---|
| AO | ACG | 25 | 140 | 200 | AO |
|  | AG |  |  |  | AO |
|  | BCG |  |  |  | AO |
|  | ABC |  |  |  | AO |
| BO | AG | 5 | 110 | 0 | BO |
|  | BCG |  |  |  | BO |
|  | BC |  |  |  | BO |
|  | ABC |  |  |  | BO |
| CO | CG | 60 | 100 | 300 | CO |
|  | ABG |  |  |  | CO |
|  | AB |  |  |  | CO |
|  | ABC |  |  |  | CO |
| AD | AG | 45 | 400 | 100 | AD |
|  | BCG |  |  |  | AD |
|  | BC |  |  |  | AD |
|  | ABC |  |  |  | AD |
| BE | BG | 45 | 250 | 400 | BE |
|  | ABG |  |  |  | BE |
|  | ACG |  |  |  | BE |
|  | ABC |  |  |  | BE |
| CF | CG | 25 | 210 | 200 | CF |
|  | ABG |  |  |  | CF |
|  | ACG |  |  |  | CF |
|  | ABC |  |  |  | CF |

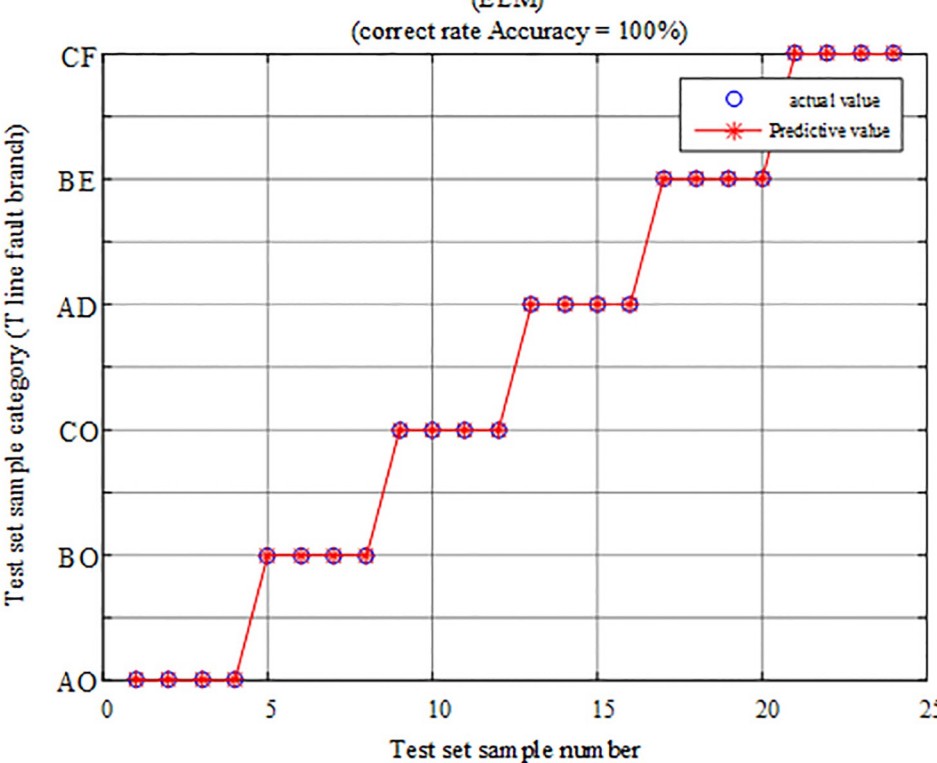

**Fig 16. Comparison of test set prediction results.**

The above chart shows that the test data in the ELM fault intelligent identification model have a correct rate of 100%. When different types of faults occur on each branch of the T-connection transmission line, the faults occurred within and outside the protection zone can be identified, and the fault branch can be accurately identified. Thus, the protection algorithm is not susceptible to type of faults.

**Analysis of different transitional resistance tests.** The fault characteristic test samples of different transitional resistances within and outside the protection zone are input into the T-connection transmission line intelligent fault identification model of limit learning machine for testing. A comparison of the prediction results is shown in Fig 16, while the simulation verification result corresponding to the fault conditions is shown in Table 4.

The above chart shows that the test sample data have a correct rate of 100% in the ELM intelligent fault identification model test, and the method can accurately identify internal and external faults and faulty branches when different transitional resistance faults occur on each branch of the T-connection transmission line. Therefore, the fault identification algorithm is not susceptible to the transitional resistance.

**Analysis of different fault distance tests.** The fault characteristic test samples of different fault distances with and outside the protection zone are input into the T-connection transmission line intelligent fault identification model of the limit learning machine for testing. The comparison of the output prediction results is shown in Fig 17, and the simulation verification results corresponding to the fault condition are reported in Table 5.

The above chart shows that the test sample data have a correct rate of 100% in the ELM intelligent fault identification model test and that the internal and external faults and faulty

**Table 4. Simulation results of different transitional resistance fault test sets.**

| Fault branch | Transition resistance / Ω | Fault initial angle/degree | Fault distance O point / km | Fault type | identification result |
|---|---|---|---|---|---|
| AO | 50 | 60 | 130 | AG | AO |
| | 100 | | | | AO |
| | 200 | | | | AO |
| | 300 | | | | AO |
| BO | 0 | 45 | 130 | ABG | BO |
| | 50 | | | | BO |
| | 100 | | | | BO |
| | 200 | | | | BO |
| CO | 50 | 25 | 80 | ACG | CO |
| | 100 | | | | CO |
| | 200 | | | | CO |
| | 300 | | | | CO |
| AD | 0 | 25 | 430 | ABG | AD |
| | 50 | | | | AD |
| | 100 | | | | AD |
| | 200 | | | | AD |
| BE | 20 | 60 | 270 | ABG | BE |
| | 100 | | | | BE |
| | 200 | | | | BE |
| | 300 | | | | BE |
| CF | 50 | 45 | 240 | ABG | CF |
| | 100 | | | | CF |
| | 200 | | | | CF |
| | 400 | | | | CF |

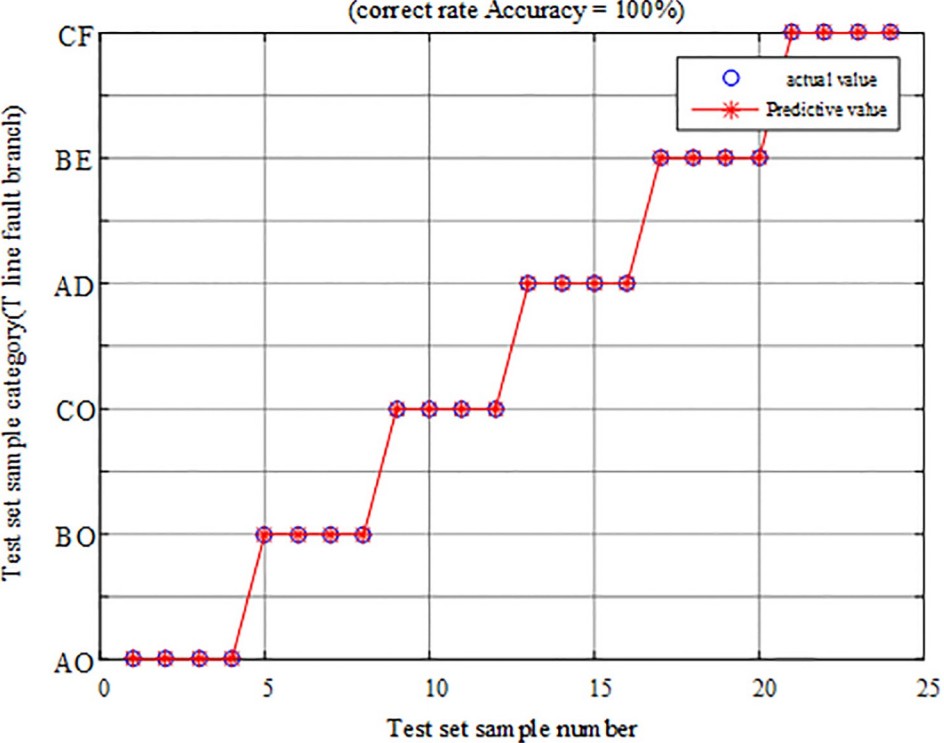

**Fig 17. Comparison of test set prediction results.**

**Table 5. Simulation results of different fault distance test sets.**

| Fault branch | Fault distance O point / km | Fault initial angle/degree | Fault type | Transition resistance / Ω | identification result |
|---|---|---|---|---|---|
| AO | 270 | 45 | ABG | 300 | AO |
| | 210 | | | | AO |
| | 160 | | | | AO |
| | 70 | | | | AO |
| BO | 170 | 45 | BCG | 50 | BO |
| | 125 | | | | BO |
| | 95 | | | | BO |
| | 55 | | | | BO |
| CO | 140 | 5 | BG | 100 | CO |
| | 110 | | | | CO |
| | 80 | | | | CO |
| | 40 | | | | CO |
| AD | 430 | 60 | BG | 200 | AD |
| | 400 | | | | AD |
| | 380 | | | | AD |
| | 340 | | | | AD |
| BE | 320 | 25 | CG | 100 | BE |
| | 280 | | | | BE |
| | 260 | | | | BE |
| | 230 | | | | BE |
| CF | 290 | 60 | ACG | 50 | CF |
| | 250 | | | | CF |
| | 210 | | | | CF |
| | 190 | | | | CF |

branches can be accurately identified at different fault distances, so the fault identification algorithm is not susceptible to fault distance.

**Analysis of the initial angle test of different faults.** The fault characteristic test samples of different fault initial angles are input into the T-connection transmission line intelligent fault identification model of the ELM for testing. A comparison of the predicted results is shown in Fig 18, and the simulation verification results corresponding to the fault condition are reported in Table 6.

The above chart shows that the test sample data are 100% correct in the test of the intelligent fault identification model of the ELM and that the internal and external faults and faulty branches can be accurately identified at different fault initial angles, so the protection algorithm is basically not susceptible to fault initial angles.

**Identification results when the fault is near point O.** The fault characteristic test sample at the fault near point O is input into the limit learning machine T-connection transmission line intelligent fault identification model for testing. A comparison of the predicted results is shown in Fig 19, and the simulation verification results corresponding to the fault condition are reported in Table 7.

The above chart shows that using the test sample data in the ELM intelligent fault identification model test results in a correct rate of 100%, so the protection algorithm can accurately identify the branch on which the fault occurs near point O of T-connection transmission line.

**Other fault condition identification results.** In order to further verify the effectiveness of the algorithm, two sets of fault conditions different from the previous ones are selected from each branch of the T- connection transmission line, and 12 sets of fault characteristic vectors are obtained by simulation, and the obtained fault characteristic test samples are input into the

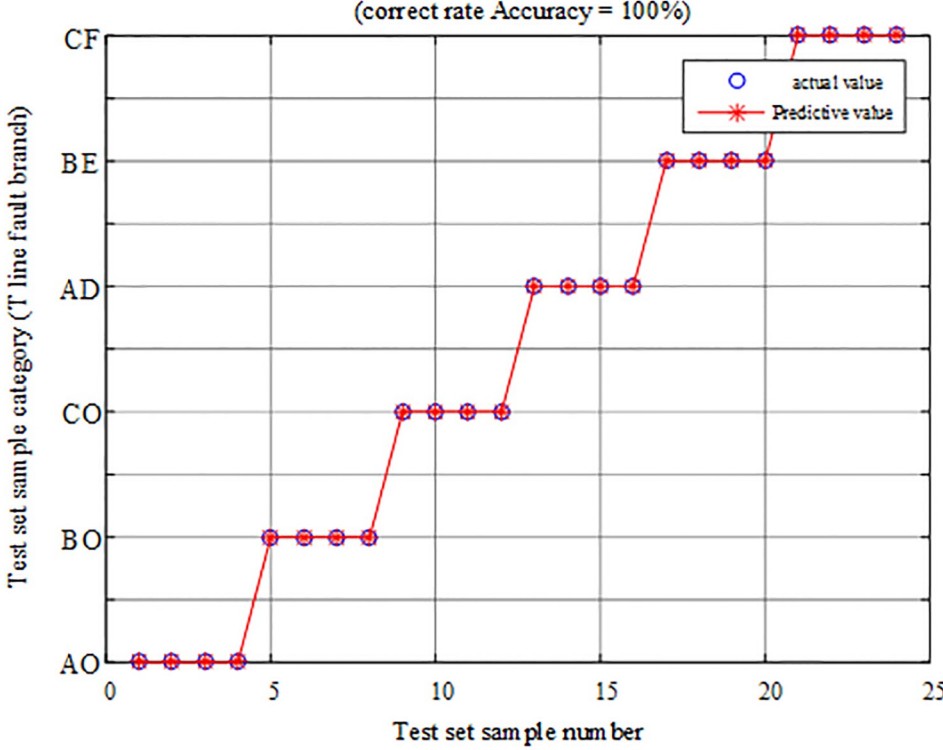

**Fig 18. Comparison of test set prediction results.**

**Table 6. Simulation results of different initial angle test sets.**

| Fault branch | Fault initial angle/degree | Fault type | Distance of fault from point O / km | Transitional resistance / Ω | identification result |
|---|---|---|---|---|---|
| AO | 5 | BG | 160 | 350 | AO |
| | 45 | | | | AO |
| | 60 | | | | AO |
| | 120 | | | | AO |
| BO | 5 | BG | 100 | 100 | BO |
| | 25 | | | | BO |
| | 60 | | | | BO |
| | 120 | | | | BO |
| CO | 5 | ABG | 50 | 250 | CO |
| | 45 | | | | CO |
| | 60 | | | | CO |
| | 120 | | | | CO |
| AD | 5 | BCG | 380 | 0 | AD |
| | 25 | | | | AD |
| | 45 | | | | AD |
| | 60 | | | | AD |
| BE | 5 | ACG | 275 | 200 | BE |
| | 45 | | | | BE |
| | 60 | | | | BE |
| | 120 | | | | BE |
| CF | 5 | BCG | 250 | 100 | CF |
| | 25 | | | | CF |
| | 45 | | | | CF |
| | 120 | | | | CF |

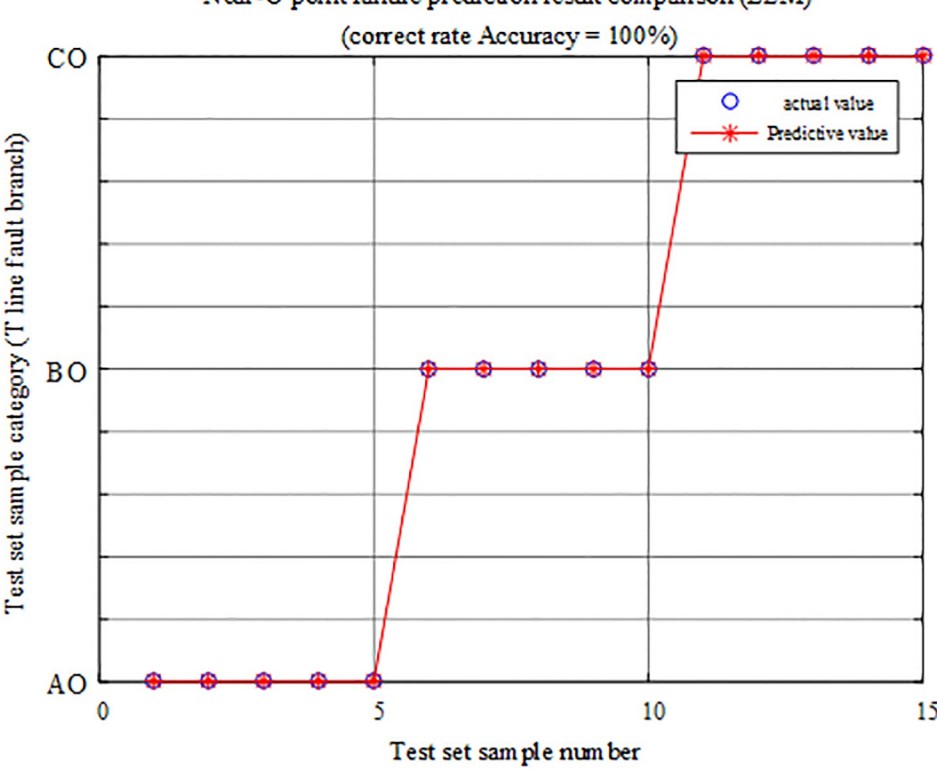

**Fig 19. Comparison of test set prediction results.**

limit learning machine T-connection transmission line intelligence fault identification model for testing, and the comparison of the predicted results is shown in Fig 20, wherein Table 8 is the simulation verification result corresponding to the fault condition.

The above diagram shows that under various fault conditions, the result of the test sample data is 100% correct in the extreme learning machine intelligent fault identification model test,

**Table 7. Simulation results of a test set in which the T-connection line near the O point fails.**

| Fault branch | Distance from point O / km | Fault type | Fault initial angle/degree | Transitional resistance / Ω | identification result |
|---|---|---|---|---|---|
| AO | 2.5 | AC | 60 | 100 | AO |
| | 2 | ABG | 120 | 50 | AO |
| | 1.5 | AG | 45 | 300 | AO |
| | 1.2 | ACG | 60 | 50 | AO |
| | 1 | BCG | 25 | 100 | AO |
| BO | 2.5 | ABG | 45 | 200 | BO |
| | 2 | AB | 5 | 100 | BO |
| | 1.5 | BCG | 120 | 200 | BO |
| | 1.2 | BG | 5 | 300 | BO |
| | 1 | ABG | 45 | 100 | BO |
| CO | 2.5 | ACG | 5 | 200 | CO |
| | 2 | BC | 45 | 50 | CO |
| | 1.5 | AG | 60 | 400 | CO |
| | 1.2 | ACG | 25 | 100 | CO |
| | 1 | BCG | 120 | 100 | CO |

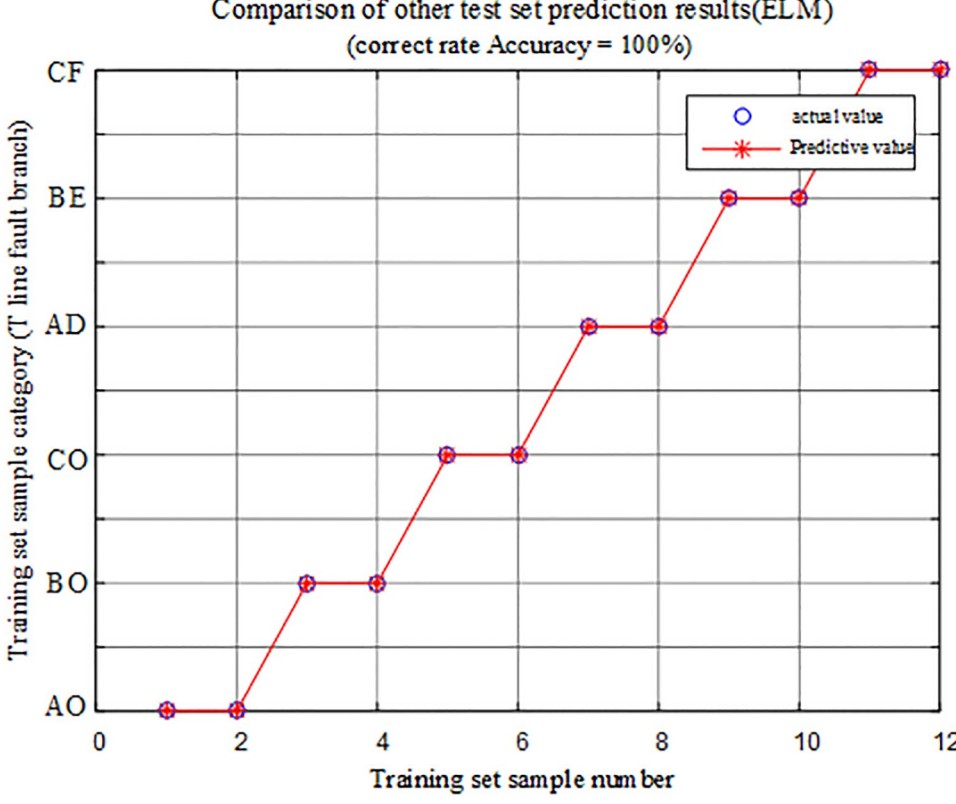

**Fig 20. Comparison of test set prediction results.**

and the protection algorithm can accurately identify the fault branch of T-connection transmission line.

## T-connection transmission line fault branch identification performance analysis

The protection speed of the traditional T-connection transmission line traveling wave is very fast, and the anti-CT saturation ability is relatively good. The impact of the capacitor current is

**Table 8. Simulation results of test set Of fault occurred near point O of the T-connection transmission line.**

| Fault branch | Fault type | Fault initial angle/degree | Distance from point O / km | Transitional resistance / Ω | identification result |
|---|---|---|---|---|---|
| AO | ABG | 5 | 210 | 200 | AO |
| | AG | 45 | 120 | 100 | AO |
| BO | AC | 100 | 120 | 50 | BO |
| | BG | 45 | 80 | 300 | BO |
| CO | CG | 60 | 75 | 100 | CO |
| | ACG | 25 | 90 | 50 | CO |
| AD | AG | 45 | 380 | 100 | AD |
| | BCG | 120 | 395 | 50 | AD |
| BE | ABG | 45 | 290 | 100 | BE |
| | AB | 5 | 275 | 200 | BE |
| CF | BCG | 100 | 230 | 100 | CF |
| | ACG | 60 | 270 | 200 | CF |

theoretically eliminated, but there have been problems with reliability. One of the main reasons is that when a small initial angle fault or a large transitional resistance fault occurs, the generated transient traveling wave signal is relatively weak, and only using the wavefront information of the initial traveling wave may cause the decrease of reliability and sensitivity of the protection. As to the T-connection transmission line fault branch identification method constructed only by using the wavefront information of the initial traveling, peak information loss may occur, and the fault branch identification can easily fail. In this paper, a performance analysis of the T-connection transmission line fault branch identification algorithm in terms of random data loss, anti-CT saturation and noise impact is performed.

**Analysis of the impact of data loss.**　In actual engineering operation, data information may be lost. To verify the performance of the algorithm in this case, take the data information loss at the frequency of 40 kHz after the implementation of S-transformation of the reverse traveling wave signal $\Delta i_{1-}(t)$ measured by the traveling wave protection unit $TR_1$ as an example. Simulation verification analysis is carried out on the fault identification algorithm.

① Analysis of the Fault Identification Algorithm for Data Loss Near the Wavefront of the Traveling Wave.

The internal branch AO and the external branch BE are selected among the branches of the T-connection transmission line, and the simulation is carried out. and 10, 15, 20, 25, 30 data information near the wavefront of the current traveling wave are set to be lost at 40 kHz after the implementation of S-transformation. 10 sets of T-connection transmission line fault characteristic test vectors are obtained. loss of 30 data points near wavefront of the traveling wave are taken as an example. Fig 21 is a waveform showing the loss of data information near the wavefront of the traveling wave when the internal branch AO is in a BCG fault at a distance of 150 km from point O. Fig 22 is a waveform showing the loss of data information of the wavefront of the traveling wave when the CG fault occurs at a distance of 280 km from point O.

The fault characteristic test sample is input into the ELM intelligent fault identification model for testing, and a comparison of the predicted results is shown in Fig 23. Table 9 presents the specific simulation verification result under the data loss of the corresponding internal branch AO and the external branch BE.

② Analysis of 2 Fault Identification Algorithms for Random Loss of Sample Points.

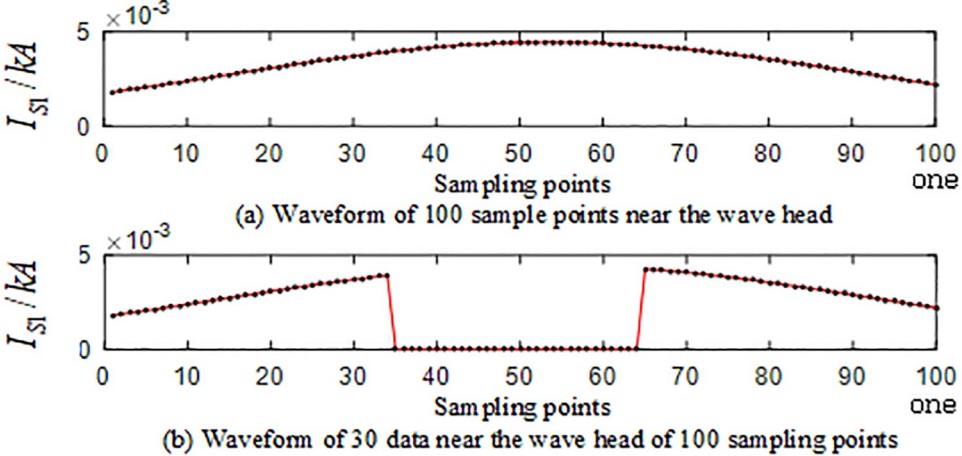

**Fig 21. Corresponding figure of data of the sampling points near the wavefront of the initial traveling wave of $TR_1$ when a fault occurs on branch AO.**

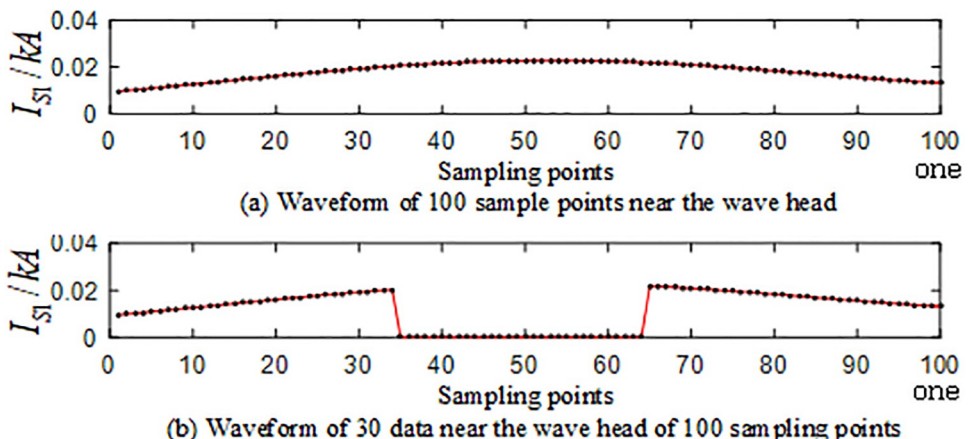

**Fig 22. Corresponding figure of data of the sampling points near the wavefront of the initial traveling wave of $TR_1$ when a fault occurs on branch BE.**

In the branches of the T-connection line, the regional branch road BO and the outer branch road CF are selected for simulation. Taking the random loss of the sampling point data of the traveling wave protection unit $TR_1$ as an example, the sampling data is randomly lost at the frequency of 40 kHz after the S-transformation. The values 10, 15, 20, 25, and 30, yield 10 sets of T-connection transmission line fault characteristic vectors. Fig 24 is a waveform related to random loss of data points of sampling points near the traveling wave head after the fault of the

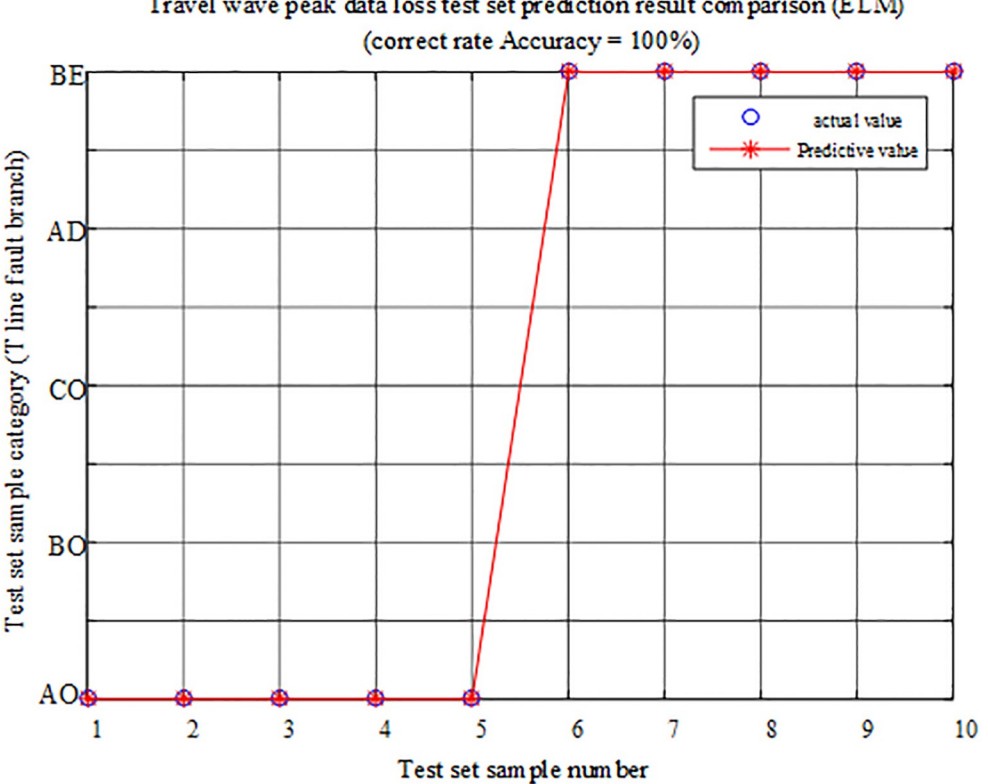

**Fig 23. Comparison of test set prediction results.**

Table 9. Simulation results of the test set under loss of AO and out-of-band branch road data in the zone.

| Fault branch | Peak data loss situation / one | Fault type | Fault initial angle/degree | Fault distance O point / km | Transition resistance / $\Omega$ | identification result |
|---|---|---|---|---|---|---|
| AO | 10 | BCG | 45 | 150 | 100 | AO |
|  | 15 |  |  |  |  | AO |
|  | 20 |  |  |  |  | AO |
|  | 25 |  |  |  |  | AO |
|  | 30 |  |  |  |  | AO |
| BE | 10 | CG | 25 | 280 | 200 | BE |
|  | 15 |  |  |  |  | BE |
|  | 20 |  |  |  |  | BE |
|  | 25 |  |  |  |  | BE |
|  | 30 |  |  |  |  | BE |

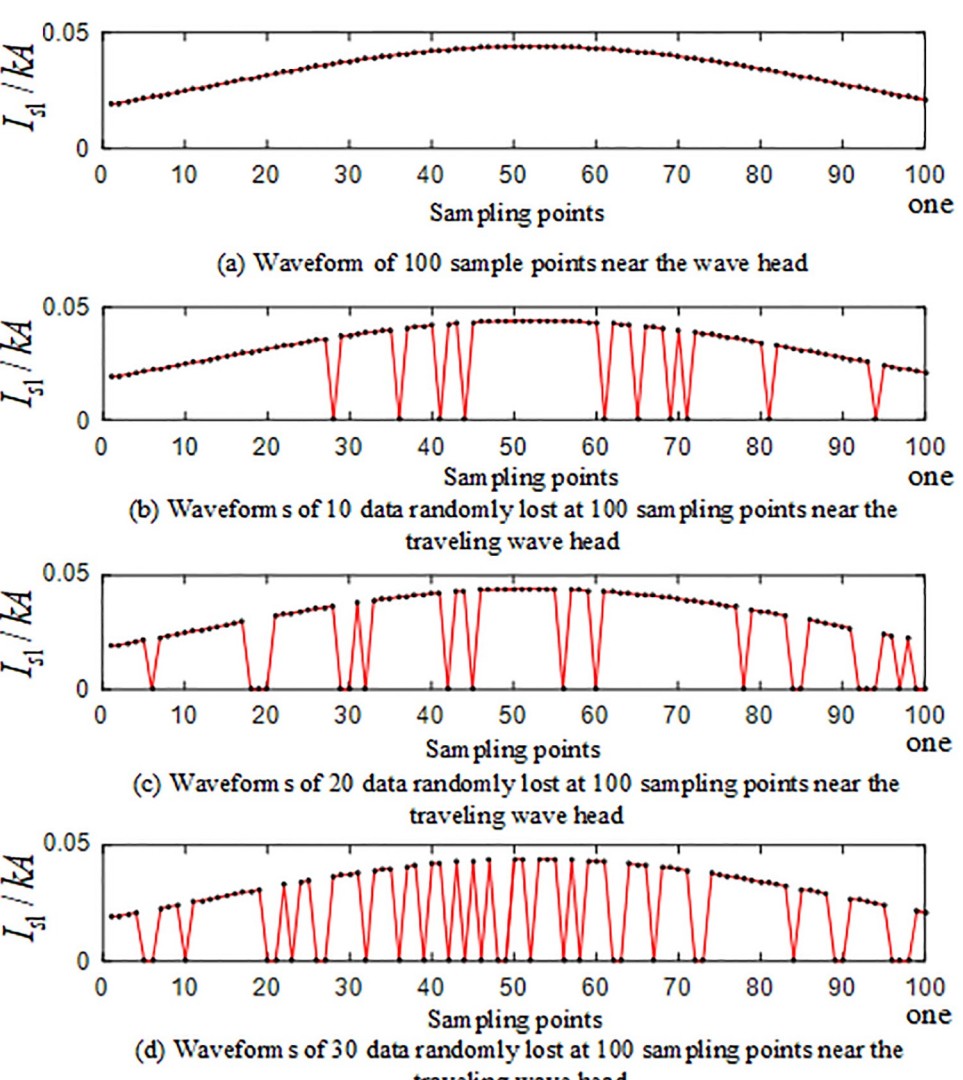

(a) Waveform of 100 sample points near the wave head

(b) Waveforms of 10 data randomly lost at 100 sampling points near the traveling wave head

(c) Waveforms of 20 data randomly lost at 100 sampling points near the traveling wave head

(d) Waveforms of 30 data randomly lost at 100 sampling points near the traveling wave head

Fig 24. Corresponding graph of random loss of 100 data sampling points near the initial traveling wave head of $TR_1$ in the case of BO branch fault.

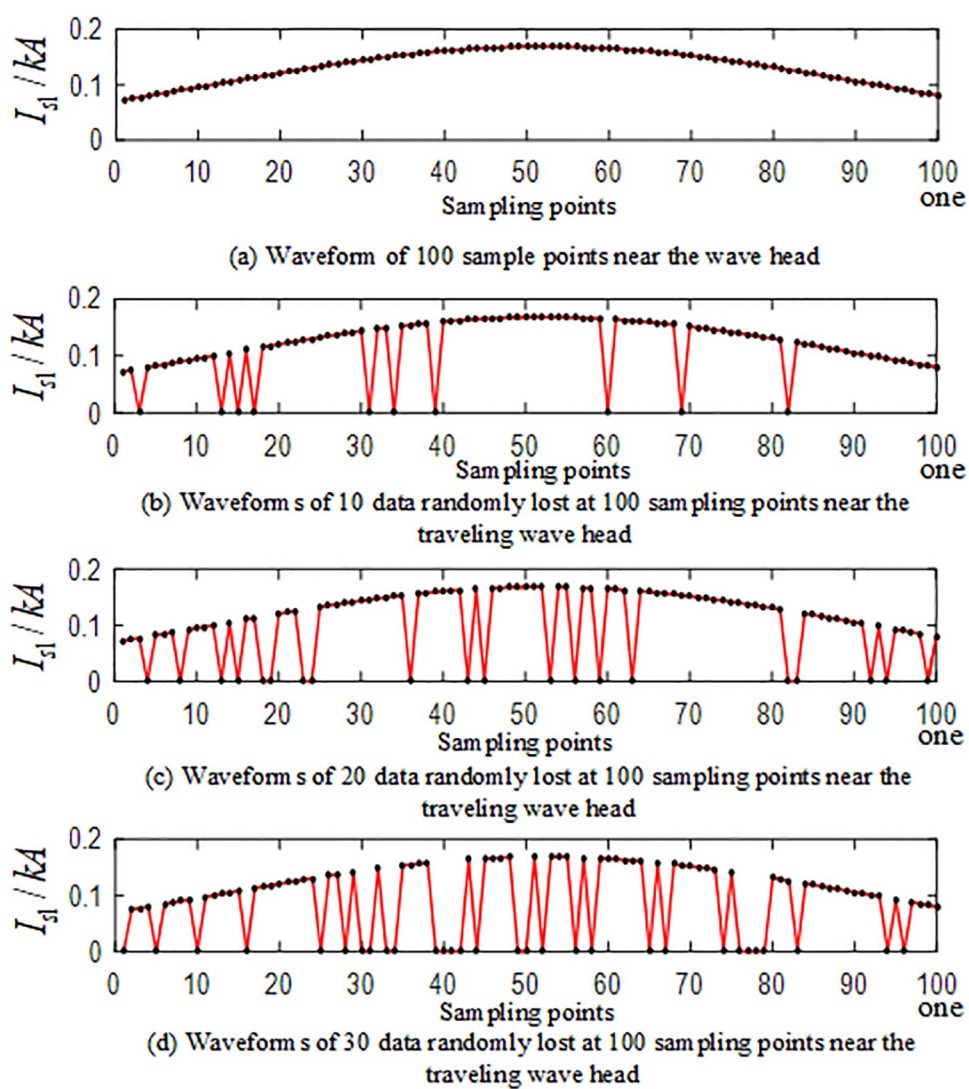

(a) Waveform of 100 sample points near the wave head

(b) Waveforms of 10 data randomly lost at 100 sampling points near the traveling wave head

(c) Waveforms of 20 data randomly lost at 100 sampling points near the traveling wave head

(d) Waveforms of 30 data randomly lost at 100 sampling points near the traveling wave head

**Fig 25. Data loss random correlation graph of 100 data sampling points near the initial traveling wave head of** $TR_1$ **in the case of CF branch fault.**

AG branch fault at 110 km from the O branch. Fig 25 is a waveform showing the random loss of data points of sampling points near the traveling wave head when the AC branch fault occurs at an outer branch CF 230 km from the O point.

The fault characteristic test sample is input into the ELM intelligent fault identification model for testing, and a comparison of the predicted results is shown in Fig 26. The specific simulation verification results under the data loss of the regional branch BO and the outer branch CF are shown in Table 10.

It can be seen from the above chart analysis that the algorithm can 100% identify the faulty branch when the external branch in the T-connection transmission line fails, and the data near the traveling wave head is lost and the sampling point data is randomly lost, which is less susceptible to data loss.

**Analysis of anti-CT saturation ability.** To verify the anti-CT saturation performance of the protection algorithm proposed in this paper, the CT saturation of each branch of the T-

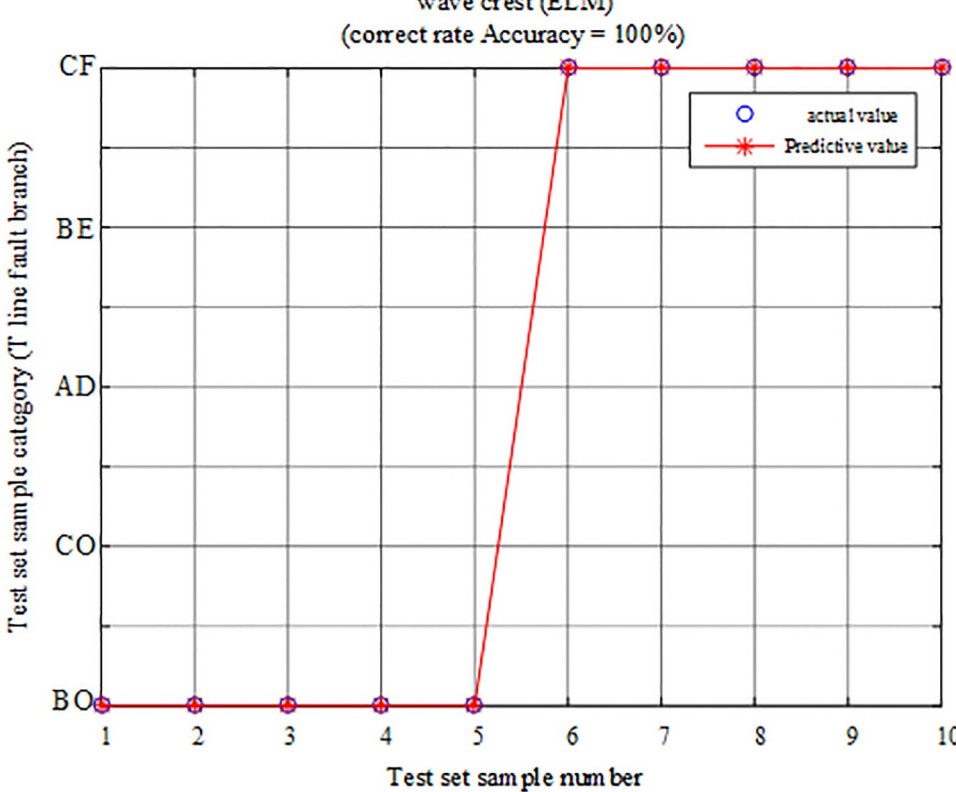

**Fig 26. Comparison of test set prediction results.**

connection transmission line is simulated and analyzed (taking the CT saturation of the branch BO in the T-connection transmission line as an example), The model uses a nonlinear time domain equivalent circuit model with better time-frequency characteristics [33].

Under the condition that the branch BO of the T-connection transmission line zone has CT saturation, a set of faults are simulated in each branch of the T-connection transmission line, and six sets of T-connection transmission line fault eigenvectors are obtained. The fault characteristic test sample matrix is input into the ELM intelligent fault identification. The test

**Table 10. Simulation results of the test set under random loss of CF data in the branch road and the outer branch road in the zone.**

| Fault branch | Random number of data lost / one | Fault type | Fault initial angle/ degree | Fault distance O point / km | Transition resistance / Ω | identification result |
|---|---|---|---|---|---|---|
| BO | 10 | AG | 5 | 110 | 100 | BO |
| | 15 | | | | | BO |
| | 20 | | | | | BO |
| | 25 | | | | | BO |
| | 30 | | | | | BO |
| CF | 10 | ACG | 45 | 230 | 50 | CF |
| | 15 | | | | | CF |
| | 20 | | | | | CF |
| | 25 | | | | | CF |
| | 30 | | | | | CF |

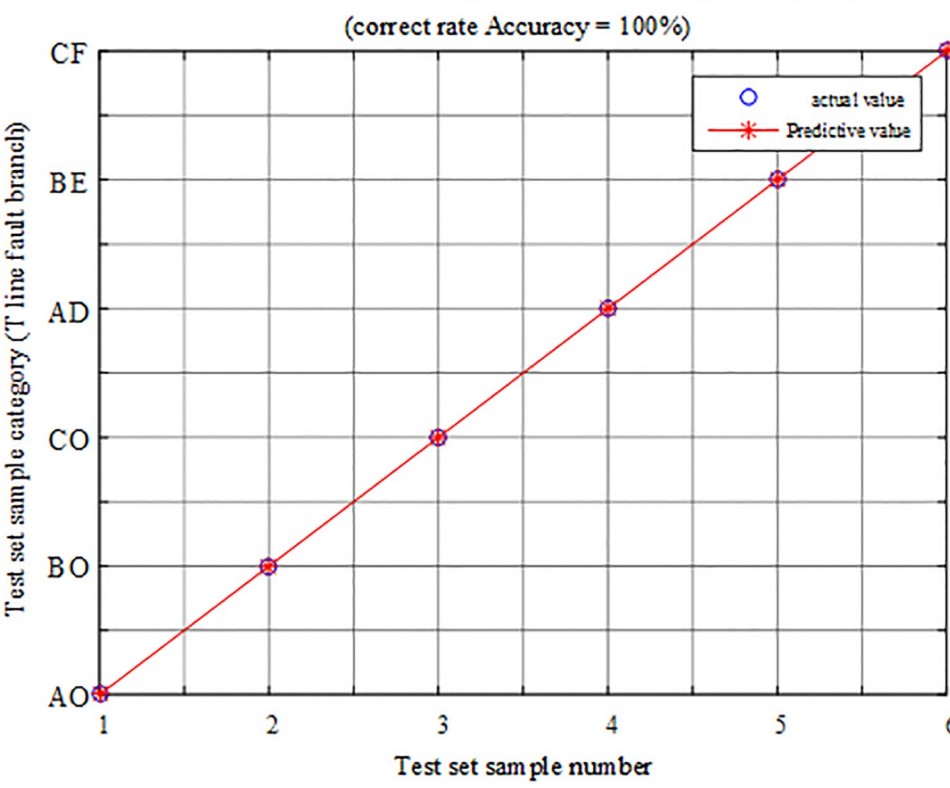

**Fig 27. Comparison of test set prediction results.**

is carried out in the model, and a comparison of the test set prediction results is shown in Fig 27. Table 11 shows the simulation test results of the CT saturation protection algorithm of the T-connection transmission line.

It can be seen from the above chart analysis that when the CT of the branch BO in the T-connection transmission line region is CT saturated, the algorithm can identify the faulty branch 100%, which is less susceptible to CT saturation.

**Noise impact analysis.**   To verify the reliability of the algorithm under the influence of noise, noise is added to the voltage and current signals measured by each traveling wave protection unit $TR_m$ ($m$ = 1, 2, 3) of the T-connection transmission line, and the signal-to-noise ratio (SBRs) is 30 dB~70 dB. Fig 28 is the current-dependent traveling wave waveform measured by the BO fault traveling wave protection unit $TR_1$ in the T-connection transmission

**Table 11.  Simulation results of the test set when the branch BO of the T-connection transmission line zone exhibits CT saturation.**

| Fault branch | Fault type | Fault initial angle/degree | Fault distance O point / km | Transition resistance / Ω | identification result |
|---|---|---|---|---|---|
| AO | ABG | 45 | 160 | 100 | AO |
| BO | BC | 60 | 100 | 100 | BO |
| CO | CG | 60 | 100 | 100 | CO |
| AD | BCG | 60 | 370 | 100 | AD |
| BE | ACG | 5 | 290 | 100 | BE |
| CF | AG | 5 | 250 | 200 | CF |

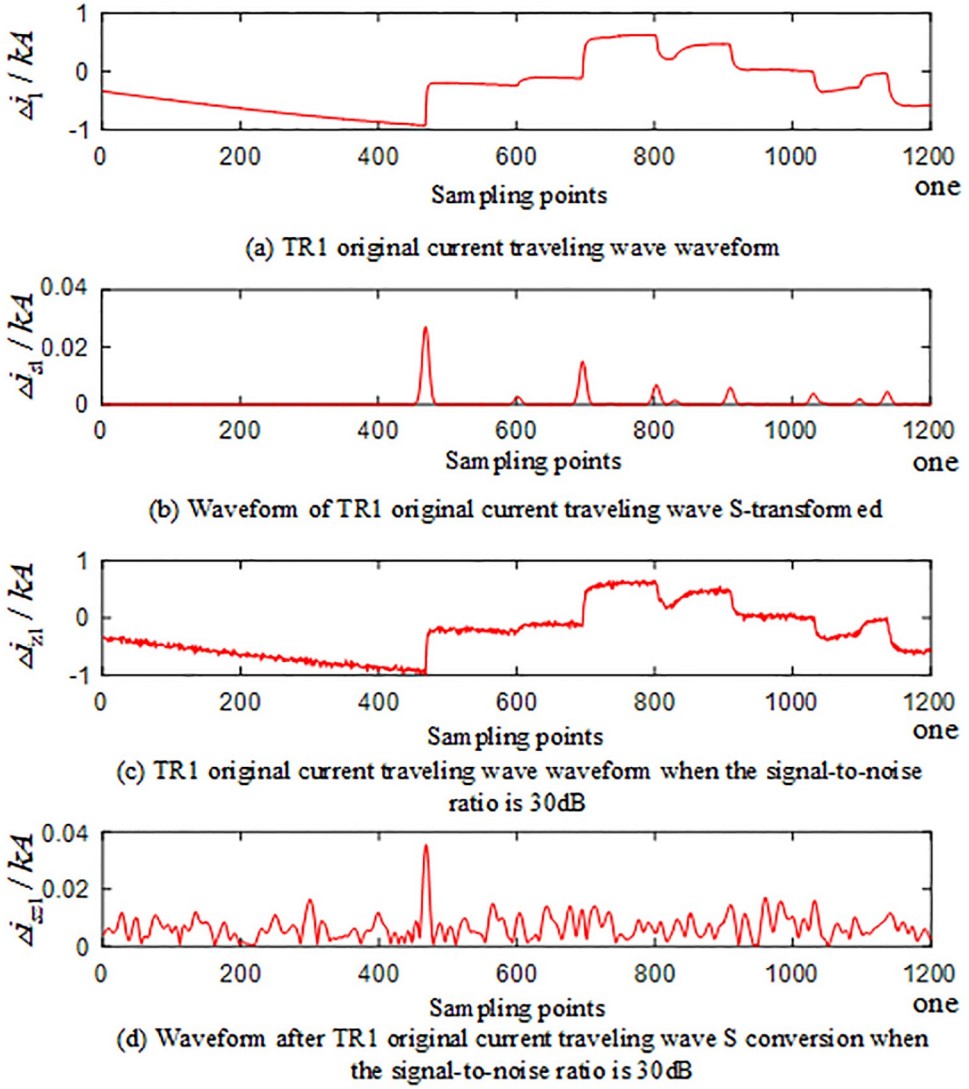

**Fig 28. Current-related waveforms measured by the branch BO fault traveling wave protection unit $TR_1$ in the T-connection transmission line zone.**

line zone, and Fig 29 is the current-dependent traveling wave waveform measured by the traveling wave protection unit $TR_1$ when the AD branch is outside the T-connection transmission line zone. (The current traveling wave measured by the traveling wave protection unit $TR_1$ is taken as an example when the signal-to-noise ratio is 30 dB and the frequency after S conversion is 40 kHz).

In the regional BO branch and the out-of-zone AD branch, a set of faults different from the training samples are selected, noise is added to the voltage and current signals, and the signal-to-noise ratio (SBRs) are 30 dB, 40 dB, 50 dB, 60 dB, and 70 dB, respectively. The simulation results show 10 sets of T-connection transmission line fault eigenvectors, and the fault characteristic test sample matrix is input into the ELM intelligent fault identification model for testing. A comparison of the predicted results is shown in Fig 30 below. Table 12 shows the regional branch BO and the outer branch. The simulation test results of the road AD under different SNR faults are provided.

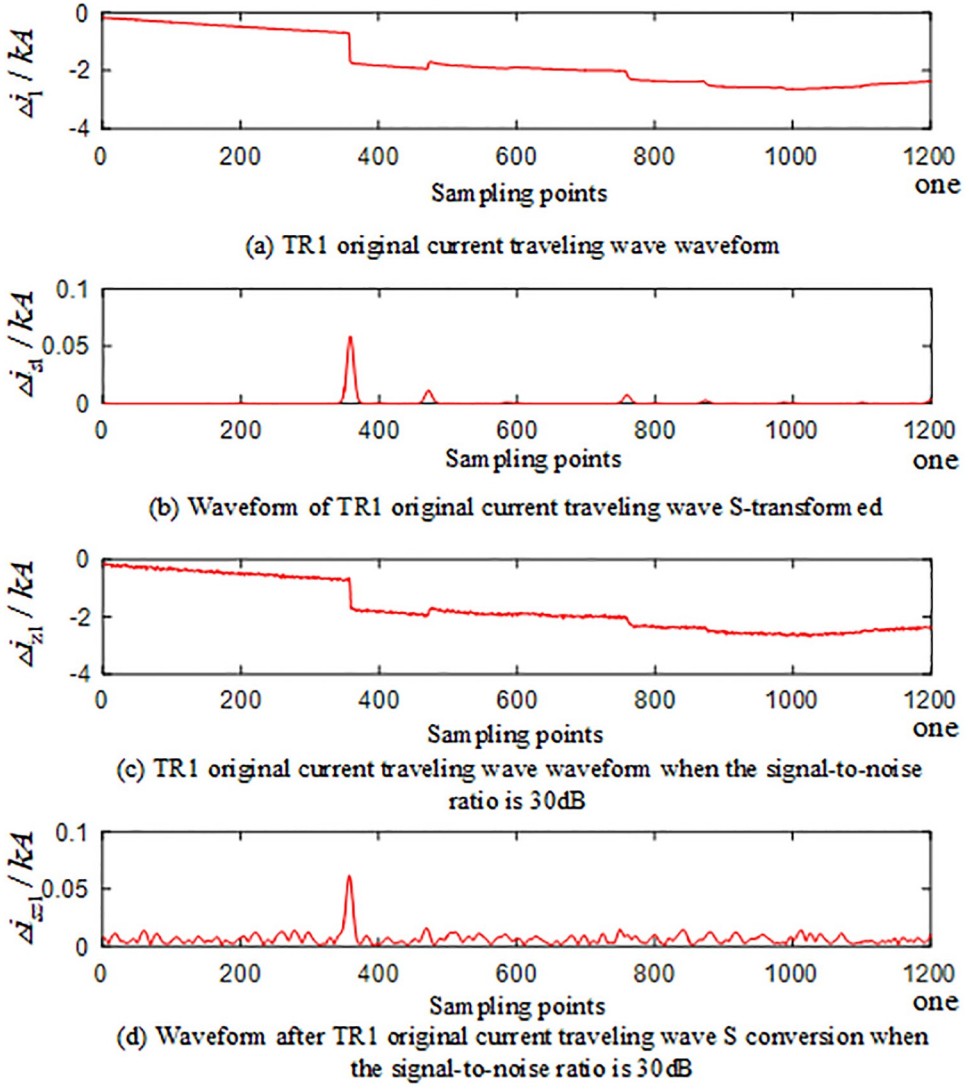

**Fig 29. Current-related waveforms measured by the AD fault traveling wave protection unit $TR_1$ of the external branch of the T-connection transmission line.**

According to the analysis of the above chart results, when the regional branch BO and the outer branch AD are under different SNR faults, the algorithm can identify the faulty branch 100%, which is only slightly susceptible to noise.

## Comparative analysis with traditional algorithms

### Motion speed analysis

At present, the traditional power frequency T-connection transmission line fault identification algorithm is widely used in the full-circumference or half-cycle Fourier algorithm. To ensure the accuracy of the calculation, the full-cycle Fourier algorithm requires a data window of 20 ms, and the half-cycle Fourier algorithm requires 10ms. The data window required by the proposed algorithm is 0.5 ms, which greatly shortens the data window length compared with the traditional power frequency T-connection transmission line fault identification algorithm.

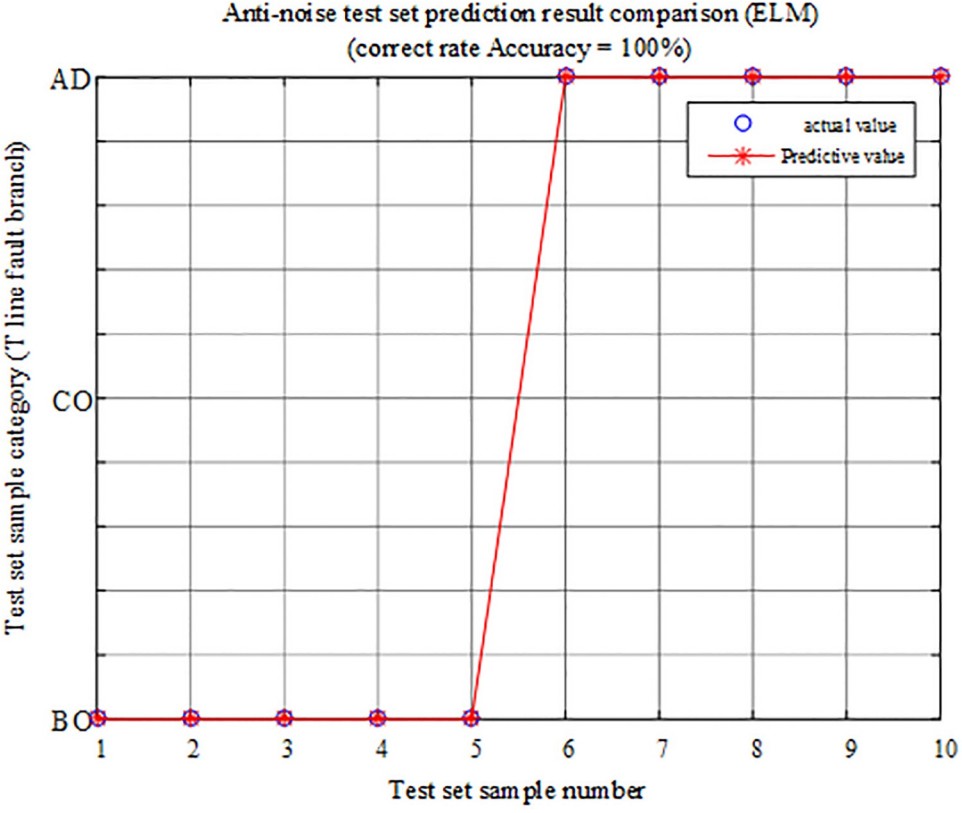

**Fig 30. Comparison of test set prediction results.**

Therefore, the proposed algorithm will have higher speed. In the traditional power frequency T connected to the line outside the fault identification algorithm.

## Fault tolerance analysis

At present, the traditional T-connection transmission line fault identification algorithm in the T-connection transmission line does not simulate the data loss, and can not verify the fault tolerance of the algorithm. In this paper, the influence of fault electrical data loss on the proposed algorithm is simulated. When the data near the traveling wave head is lost and the sampling point data is randomly lost, the simulation results show that the algorithm can identify the fault quickly and accurately.

**Table 12. Simulation results of test set for branch BO and out-of-band branch AD in T-connection transmission zone under different SNR faults.**

| Fault branch | SNR/(dB) | Fault type | Fault initial angle/degree | Fault distance O point / km | Transition resistance / Ω | identification result |
|---|---|---|---|---|---|---|
| BO | 30 | ABG | 45 | 130 | 50 | BO |
| | 40 | | | | | BO |
| | 50 | | | | | BO |
| | 60 | | | | | BO |
| | 70 | | | | | BO |
| AD | 30 | ABG | 25 | 430 | 50 | AD |
| | 40 | | | | | AD |
| | 50 | | | | | AD |
| | 60 | | | | | AD |
| | 70 | | | | | AD |

## Identification accuracy analysis

At present, the traditional T-connection transmission line fault identification algorithm can identify the faults in the T-connection transmission line, but it can not identify the specific branch of the fault in the zone and outside, and does not analyze the influence of noise on the accuracy of the algorithm identification. Moreover, the selection of the braking coefficient in the traditional current differential protection criterion based on the fault component is especially important for the sensitivity and reliability of fault identification in the T-connection transmission line. For example, for the protection criteria mentioned in [7], when braking if the coefficient is too large, the sensitivity of the internal fault action is not guaranteed. When the brake coefficient is selected too small, the reliability of the external fault brake is bound to decrease. The new method of T-line fault identification based on S-transform energy entropy and extreme learning machine proposed in this paper, using the advantages of high learning efficiency and generalization ability of extreme learning machine to identify fault branch of T-connection line. In the process, not only can the faults in the area and outside be accurately and effectively identified, but also the specific branch of the fault can be identified.

Compared with the traditional fault identification algorithm, this paper proposes a new method of T-connection transmission line fault identification based on S-transform energy entropy and an extreme learning machine, which utilizes the advantages of high learning efficiency and generalization ability of the extreme learning machine. Road identification, in the process of fault identification, not only can accurately identify the faults inside and outside the zone, but can also identify the specific branch of the fault. And in the simulation experiment under the influence of 30–70 db of noise, the faulty branch can be identified quickly and accurately.

## Conclusions

In this paper, a new T-connection transmission line fault identification method based on current back wave multiscale S-transform energy entropy and an extreme learning machine is proposed. The characteristics of the reverse traveling wave in the T-connection transmission line internal fault and external fault are analyzed. The experiment verifies the feasibility of the fault identification method. The theoretical and simulation results show that:

1. The algorithm identifies the faulty branch in the T-connection transmission line by establishing a good T-connection transmission line intelligent fault identification model. In the simulation analysis under various working conditions, the fault can be quickly and accurately identified, and the transition is basically overcome. The influence of factors such as resistance and initial angle of failure is analyzed.

2. The algorithm can correctly identify the faulty branch under the influence of data loss, CT saturation and noise.

3. Compared with the traditional T-connection transmission line fault identification algorithm, the proposed algorithm exhibits a better performance in terms of motion speed, fault tolerance and fault identification accuracy.

## Supporting information

**S1 Fig. 500kV T-connected transmission line Model built by PSCAD.**
(TIFF)

**S1 Table. Simulation results of different fault type test sets.**
(DOCX)

**S2 Table. Simulation results of different transitional resistance fault test sets.**
(DOCX)

**S3 Table. Simulation results of different fault distance test sets.**
(DOCX)

**S4 Table. Simulation results of different initial angle test sets.**
(DOCX)

**S5 Table. Simulation results of a test set in which the T-connection line near the O point fails.**
(DOCX)

**S6 Table. Simulation results of the test set under loss of AO and out-of-band branch road data in the zone.**
(DOCX)

**S7 Table. Simulation results of the test set under random loss of CF data in the branch road and the outer branch road in the zone.**
(DOCX)

**S8 Table. Simulation results of the test set when the branch BO of the T-connection transmission line zone exhibits CT saturation.**
(DOCX)

**S9 Table. Simulation results of test set for branch BO and out-of-band branch AD in T-connection transmission zone under different SNR faults.**
(DOCX)

**S10 Table. The partial data obtained from Fig 4 is as follows.**
(DOCX)

**S11 Table. The partial data obtained from Fig 5 is as follows.**
(DOCX)

**S12 Table. The partial data obtained from Fig 6 is as follows.**
(DOCX)

**S13 Table. The partial data obtained from Fig 7 is as follows.**
(DOCX)

**S14 Table. The partial data obtained from Fig 8 is as follows.**
(DOCX)

**S15 Table. The partial data obtained from Fig 9 is as follows.**
(DOCX)

**S16 Table. The data obtained from Fig 21 is as follows.**
(DOCX)

**S17 Table. The data obtained from Fig 22 is as follows.**
(DOCX)

**S18 Table. The data obtained from Fig 24 is as follows.**
(DOCX)

**S19 Table. The data obtained from Fig 25 is as follows.**
(DOCX)

**S20 Table. The partial data obtained from Fig 28 is as follows.**
(DOCX)

**S21 Table. The partial data obtained from Fig 29 is as follows.**
(DOCX)

## Acknowledgments

This research was supported by National Natural Science Foundation of China (Grant Nos. 11705122); The Project of Sichuan provincial science and Technology Department (Grant No. 2017JY0338. 2019YJ0477, 2018GZDZX0043); The artificial intelligence key laboratory of Sichuan province Foundation (2017RYY02); Sichuan University of Science and Engineering talent introduction project (2017RCL53); Enterprise informatization and Internet of things measurement and control technology key laboratory project of Sichuan provincial university (2018WZY01); The Project of Sichuan Provincial Academician (Expert) workstation of Sichuan University of Science and Engineering (2018YSGZZ04).

## Author Contributions

**Data curation:** Jie Yang.

**Formal analysis:** Leilei Chen, Qiaomei Wang.

**Funding acquisition:** Hao Wu.

**Methodology:** Jie Yang.

**Project administration:** Hao Wu.

**Software:** Leilei Chen.

**Visualization:** Qiaomei Wang.

**Writing – original draft:** Hao Wu.

**Writing – review & editing:** Jie Yang.

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
