## [Decision Letter · Decision Letter 0]

18 Jun 2019

PONE-D-19-14484

A New Method for Identifying a Fault in T-Connected Lines Based on MultiScale S-Transform Energy Entropy and an Extreme Learning Machine

PLOS ONE

Dear Dr Wu,

Thank you for submitting your manuscript to PLOS ONE. After careful consideration, we feel that it has merit but does not fully meet PLOS ONE’s publication criteria as it currently stands. Therefore, we invite you to submit a revised version of the manuscript that addresses the points raised during the review process.

Please carefully revised the manuscript by considering the reviewer's comments.

We would appreciate receiving your revised manuscript by Aug 02 2019 11:59PM. To enhance the reproducibility of your results, we recommend that if applicable you deposit your laboratory protocols in protocols.io, where a protocol can be assigned its own identifier (DOI) such that it can be cited independently in the future. For instructions see: http://journals.plos.org/plosone/s/submission-guidelines#loc-laboratory-protocols

We look forward to receiving your revised manuscript.

Kind regards,

Jie Zhang

Academic Editor

PLOS ONE

Journal Requirements:

Reviewers' comments:

Reviewer's Responses to Questions

**Comments to the Author**

1. Is the manuscript technically sound, and do the data support the conclusions?

Reviewer #1: Partly

2. Has the statistical analysis been performed appropriately and rigorously? 

Reviewer #1: Yes

3. Have the authors made all data underlying the findings in their manuscript fully available?

Reviewer #1: Yes

4. Is the manuscript presented in an intelligible fashion and written in standard English?

Reviewer #1: No

5. Review Comments to the Author

Reviewer #1: The paper is not written with technical English and is really hard to follow. It is just like translated from some sorts of software. Please proofread the whole paper carefully to correct all grammar mistakes. Though the paper is with good novelty, I cannot recommend the paper for publication unless the English is improved.

Introduction part is poor at criticizing literature works. It is not clear why the authors choose the current back wave multiscale S-transform energy entropy and an extreme learning machine

Inductance and capacitance of the unit length line are not reflected in Figure 1 and the resistance is ignored

Figure 4. Figure 5. Figure 6 and so on without the abscissa.

The author proposed the single-hidden layer feedforward neural network (SLFN) learning algorithm，but the following paper is about ELM, What is the connection between SLFN and ELM?

6. PLOS authors have the option to publish the peer review history of their article (what does this mean?). If published, this will include your full peer review and any attached files.

Reviewer #1: No

---

## [Author Response · Author response to Decision Letter 0]

11 Jul 2019

Dear Editor:

The article " A New Method for Identifying a Fault in T-Connected Lines Based on MultiScale S-Transform Energy Entropy and an Extreme Learning Machine" has been revised in strict accordance with the comments of the review experts. The specific changes are as follows, please edit your review.

Comment 1：

1、The paper is not written with technical English and is really hard to follow. It is just like translated from some sorts of software. Please proofread the whole paper carefully to correct all grammar mistakes. Though the paper is with good novelty, I cannot recommend the paper for publication unless the English is improved.

Modify the description：

The article has been revised in English as required.

Comment 2：

2、Introduction part is poor at criticizing literature works. It is not clear why the authors choose the current back wave multiscale S-transform energy entropy and an extreme learning machine.

Modify the description：

The article has revised the introduction part according to the requirements, pointed out the shortcomings of the traditional T-connection transmission line fault identification algorithm, and summarized the problems existing in the traditional T-connection transmission line fault identification algorithm. The details are as follows:

Introduction

With the continuous development of the social economy, the complexity of the power grid has gradually increased. Considering the investment savings and other restrictions on objective conditions, T-connection transmission lines are increasingly appearing in high-voltage and ultrahigh-voltage power networks due to the uniqueness of their modes of connection. However, these lines are often accompanied by large power plants and systems, whose transmission lines have high transmission power and heavy load. When the line fails, it may cause large-scale blackouts. Therefore, when a fault occurs, it is required to be able to quickly and accurately identify the fault [1-5].

At present, researches on T-connection transmission line fault identification domestic and foreign scholar have carried out are mainly based on the voltage, current and transmission line distribution parameter model. In reference[6],faults occurred within and outside of the protection zone are identified by using the ratio of the phasor sum of T-connection line three-terminal voltage fault component and the phasor sum of the current fault component. Reference[7] uses the sum of the three-terminal current fault components of the T-connection transmission line and the vector difference between the maximum current in the three-terminal current fault components and the sum of the currents of other two terminals to establish a criterion to identify internal and external faults, but the selection of the braking coefficient in the criterion will have an impact on the sensitivity and reliability of fault identification . Aiming at addressing the problems in reference[7], reference[8] utilized the maximum current in the three-terminal fault current components of the T-connection transmission line with the other two ends and the remaining string angles to establish a criterion to identify internal and external faults, but did not analyze the performance of the algorithm under the influence of noise. Based on the criteria presented in references [7-8], reference [9] establishes a comprehensive criterion to identify faults occurred in the photovoltaic T-connection high voltage distribution network. However, in reference [9], data loss is not discussed in the process of simulation analysis of the algorithm. In reference[10], by using the information of the voltage, current and transmission line positive sequence impedance parameters of each side of the T-connection transmission line, the T-connection voltage is obtained from each side, and then the faulty branch is identified by using the obtained T-connection voltage amplitude information. Reference[11] provides the voltage and current signals measured by the T-connection transmission line protection terminal to the second-order Taylor-Kalman-Fourier (T2KF) filter to estimate the instantaneous values of the voltage and current signals, and then calculates the positive sequence impedance to identify the faulty section. In reference[12], the positive sequence voltage at the T-connection is calculated at the three ends of the T-connection transmission line, and internal and external faults are identified by comparing the maximum amplitude of the T-connection positive sequence voltage superposition component with the maximum amplitude of the three-terminal positive sequence voltage superposition component. In reference[13], the maximum value of the T-connection positive sequence superimposed voltage calculated by the three ends of the T-connection transmission line is used to determine whether the line is faulty, and the phase difference between the positive sequence superimposed voltage and the current at a particular terminal is then used to identify internal and external faults. Reference [14] uses the voltage amplitude difference and measured impedance characteristics of the three sides of the T-connection transmission line to establish the main criterion of the integrated voltage amplitude difference, and with the combination of adaptive distance auxiliary criterion, internal and external faults can be identified. However, the performance of the algorithm has not been simulated. In [15-16], wavelet transform is applied to T-connection transmission line fault identification, but the high-frequency noise signal will affect the identification of faults occurred on T-connection transmission line. In reference[15], the bior3.1 wavelet is used to decompose the three-terminal raw current signal of the T-connection transmission line, the decomposed signal is reconstructed, and then the reconstructed signal is used to calculate the running current and the suppressing current of each phase. Finally, internal and external faults are identified by comparing the relationship between the three-phase corresponding phase running current and the suppressing current. Reference [16] distinguishes internal and external faults faults by comparing the polarity of the fault current detected by the Haar wavelet function at each end of the T-connection transmission line. The fault identification algorithm in reference [17] and [18] is mainly based on the distribution parameters of the T-connection transmission line. Reference [17] discriminates internal and external faults by comparing the exponential sum derived from the model of the transmission line, while reference [18] derives the ranging function based on the distribution parameter model of the transmission line, and uses the phase information at both ends of each branch of the ranging function to determine the branch where the fault is located..

In the traditional fault identification research of T-connection transmission line, the T-connection fault identification algorithm can only identify internal and external faults, but fail to identify the specific branch on which the fault occurred, and the fault identification accuracies in some algorithms are susceptible to other variables. In terms of fault tolerance, the traditional T-connection transmission line fault identification algorithm does not simulate the fault data loss, and cannot verify the fault tolerance of the algorithm. In terms of noise impact, the traditional T-connection transmission line fault identification algorithm does not conduct an in-depth study on it. In order to overcome the shortcomings that the traditional T-connection transmission line fault identification algorithm have in identification precision, accuracy, fault tolerance and effects from noise, this paper further studies the T-connection transmission line fault identification algorithm.

In recent years, S-transformation and information entropy theory have frequently been applied in power systems [19-21]. Reference [22] uses the S-transformed sample entropy ratio of the fault current traveling wave at both ends of the transmission line within a period of time after the fault occurred to identify internal and external faults . Reference [23] established the criterion based on the energy entropy change characteristics obtained by the reverse traveling wave S transform after the faults occurred on each associated line of the busbar to identify internal and external faults.

Based on the theory of directional traveling wave and information entropy expounded in reference [21-23] and with the application of S transform in power system [22-23], this paper proposes a new fault identification method for T-connection transmission line based on the multi-scale S transform energy entropy and limit learning machine of current reverse traveling wave. On the basis of S transformation of fault reverse traveling waves at each end of T-connection transmission line, energy entropy of the reverse traveling waves at 8 different frequencies is calculated to form a sample set of fault characteristic vectors of T-connection transmission line. Combined with the limit learning machine fault intelligent identification model, training and testing are conducted to identify fault branches of T-connection transmission line. Simulation results show that the proposed algorithm can accurately identify the T-connection transmission line branch where the internal or external fault is located under various operating conditions.

Comment 3：

3、Inductance and capacitance of the unit length line are not reflected in Figure 1 and the resistance is ignored

Modify the description：

The article has supplemented the T-connection transmission line parameters in the simulation and experimental section of the fifth section as required. The details are as follows:

Simulation and experiments

The PSCAD/EMTDC electromagnetic transient simulation software is used to establish a 500kV T-connection transmission line simulation model shown in Fig 12. The model adopts the frequency dependent distribution parameter model that can accurately reflect harmonic and transient responses. TOWER:3H5 pole tower is selected as the type of the line. The configuration of the power transmission line is shown in figure 12 below. The parameters of the transmission line are shown in Table 1 and Table 2 below. The simulation sampling frequency is 200 kHz, and the length of each branch is AO = 300 km, BO = 200 km, CO = 150 km, AD = 170 km, BE = 150 km, CF = 180 km, respectively. The current reverse traveling wave data corresponding to 5, 10, 15, 20, 25, 30, 35 and 40 kHz of the traveling wave protection unit after the implementation of S-transformation are selected in order to calculate the reverse traveling wave energy entropy of each frequency. The energy entropy vector of multi-scale reverse traveling wave is constructed. The energy entropy vector of multi-scale reverse traveling wave of three traveling wave protection units is combined into a T-connection transmission line fault eigenvector W to reflect the feature of the fault occurred on the branch of the T-connection transmission line. The sample data of the extreme learning machine is constructed, where. 

Fig 12. power line configuration

Table 1. Transmission line parameter I

Type of the Line Parameter numerical value

Phase line Wire radius/m 0.0203454

 DC Resistor/(Ω/km) 0.03206

Ground wire Wire radius/m 0.0055245

 DC Resistor/(Ω/km) 2.8645

Table 2. Transmission line parameters II

 Resistance R(Ω/km) Reactance X(Ω/km) Conductance G(s/km) Senator B(s/km) Capacitance C(μF/km)

POS 0.0346755486 0.423365555 0.0000001 0.00000272598288 0.0135

ZERO 0.30002296 1.1426412 0.0000001 0.000193555082 0.0092

Comment 4：

4、Figure 4. Figure 5. Figure 6 and so on without the abscissa.

Modify the description：

The article has been modified as required, adding the abscissa to the corresponding graph.

Comment 5：

5、The author proposed the single-hidden layer feedforward neural network (SLFN) learning algorithm，but the following paper is about ELM, What is the connection between SLFN and ELM?

Modify the description：

Feedforward neural network is one of the artificial neural networks[29]. In this kind of neural network, each neuron starts from the input layer, receives the first input, and inputs to the next level until the output layer. There is no feedback throughout the network, and a directed acyclic graph can be used. Feedforward neural network is the earliest proposed artificial neural network and the simplest type of artificial neural network. According to the number of layers of the feedforward neural network, it can be divided into a single layer feedforward neural network and a multilayer feedforward neural network. In this paper, the extreme learning machine neural network is used to train the fault sample data. It is a typical single hidden layer feedforward neural network. The network consists of an input layer, an implicit layer, and an output layer. The neurons of the input layer and the hidden layer, and the neurons of the the hidden layer and the output layer are fully connected. 

The details are as follows:

Feedforward neural network is one of the artificial neural networks[29]. In this kind of neural network, each neuron starts from the input layer, receives the first input, and inputs to the next level until the output layer. There is no feedback throughout the network, and a directed acyclic graph can be used. Feedforward neural network is the earliest proposed artificial neural network and the simplest type of artificial neural network. According to the number of layers of the feedforward neural network, it can be divided into a single layer feedforward neural network and a multilayer feedforward neural network. Among them, common feedforward neural networks include BP neural network[29], radial basis function (RBF) neural network [29] and extreme learning machine (ELM) neural network [30].

An ELM is an easy-to-use and effective single-hidden layer feedforward neural network (SLFN) learning algorithm [30]. The network consists of an input layer, an implicit layer, and an output layer. The neurons of the input layer and the hidden layer, and the neurons of the the hidden layer and the output layer are fully connected. Among them, the input layer has n neurons, corresponding to n input variables; the hidden layer has 1 neuron; the output layer has m neurons, corresponding to m output variables. Fig 10 is a single hidden layer ELM network structure.

Fig 10． ELM network structure

ELM only needs to set the number of hidden layer neurons in the network. It does not need to adjust the input weight of the network and the bias of the hidden element during the execution of the algorithm. Compared with the traditional neural network [31-32], it changes the idea that BP neural network should be based on the gradient descent learning and does not need to update the network parameters iteratively. It changes the feature that the learning performance of SVM depends too much on parameter adjustment, and has the advantages of fast learning speed and good generalization performance, and only produces the unique optimal solution.

New References:

29. Mingjie Chen, Jinren Ni, An Xue, et al. Comparative Study on Typical Feed-Forward ANNs for Tidal Simulation[J]. Journal of Sediment Research , ,2003(05):41-48.

30. Xueqing Zhang, Jun Liang, Xi Zhang, et al. Combined Model for Ultra Short-term Wind Power Prediction Based on Sample Entropy and Extreme Learning Machine[J]. Proceedings of the CSEE ,2013,33(25):33-40+8.

add content:

1. In terms of simulation, this paper adds 5.3.6 other fault identification results.

Modify the description：

Other fault condition identification results

In order to further verify the effectiveness of the algorithm, two sets of fault conditions different from the previous ones are selected from each branch of the T- connection transmission line, and 12 sets of fault characteristic vectors are obtained by simulation, and the obtained fault characteristic test samples are input into the limit learning machine T-connection transmission line intelligence fault identification model for testing, and the comparison of the predicted results is shown in Fig 20, wherein Table 8 is the simulation verification result corresponding to the fault condition.

Fig 20. Comparison of test set prediction results

Table 8. Simulation results of test set Of fault occurred near point O of the T-connection transmission line 

Fault branch Fault type Fault initial angle/degree Distance from point O / km Transitional resistance / Ω Branch identification result

AO ABG 5 210 200 AO

 AG 45 120 100 AO

BO AC 100 120 50 BO

 BG 45 80 300 BO

CO CG 60 75 100 CO

 ACG 25 90 50 CO

AD AG 45 380 100 AD

 BCG 120 395 50 AD

BE ABG 45 290 100 BE

 AB 5 275 200 BE

CF BCG 100 230 100 CF

 ACG 60 270 200 CF

The above diagram shows that under various fault conditions, the result of the test sample data is 100% correct in the extreme learning machine intelligent fault identification model test, and the protection algorithm can accurately identify the fault branch of T-connection transmission line.

---

## [Decision Letter · Decision Letter 1]

25 Jul 2019

A New Method for Identifying a Fault in T-Connected Lines Based on MultiScale S-Transform Energy Entropy and an Extreme Learning Machine

PONE-D-19-14484R1

Dear Dr. Wu,

We are pleased to inform you that your manuscript has been judged scientifically suitable for publication and will be formally accepted for publication once it complies with all outstanding technical requirements.

With kind regards,

Jie Zhang

Academic Editor

PLOS ONE

Additional Editor Comments (optional):

Reviewers' comments:

Reviewer's Responses to Questions

**Comments to the Author**

1. If the authors have adequately addressed your comments raised in a previous round of review and you feel that this manuscript is now acceptable for publication, you may indicate that here to bypass the “Comments to the Author” section, enter your conflict of interest statement in the “Confidential to Editor” section, and submit your "Accept" recommendation.

Reviewer #1: All comments have been addressed

2. Is the manuscript technically sound, and do the data support the conclusions?

Reviewer #1: Yes

3. Has the statistical analysis been performed appropriately and rigorously? 

Reviewer #1: Yes

4. Have the authors made all data underlying the findings in their manuscript fully available?

Reviewer #1: Yes

5. Is the manuscript presented in an intelligible fashion and written in standard English?

Reviewer #1: Yes

6. Review Comments to the Author

Reviewer #1: (No Response)

7. PLOS authors have the option to publish the peer review history of their article (what does this mean?). If published, this will include your full peer review and any attached files.

Reviewer #1: No

---

## [Editor Report · Acceptance letter]

1 Aug 2019

PONE-D-19-14484R1 

A New Method for Identifying a Fault in T-Connected Lines Based on MultiScale S-Transform Energy Entropy and an Extreme Learning Machine 

Dear Dr. Wu:

I am pleased to inform you that your manuscript has been deemed suitable for publication in PLOS ONE. Congratulations! Your manuscript is now with our production department. 

With kind regards,

on behalf of

Dr. Jie Zhang 

Academic Editor

PLOS ONE